# Lasting impact of winds on Arctic sea ice through the ocean's memory

Qiang Wang[1,2], Sergey Danilov[1,3,4], Longjiang Mu[5], Dmitry Sidorenko[1], and Claudia Wekerle[1]

[1]Alfred-Wegener-Institut Helmholtz-Zentrum für Polar- und Meeresforschung (AWI), Bremerhaven, Germany
[2]Laboratory for Regional Oceanography and Numerical Modeling, Pilot National Laboratory for Marine Science and Technology, Qingdao, China
[3]Department of Mathematics and Logistics, Jacobs University, Bremen, Germany
[4]A. M. Obukhov Institute of Atmospheric Physics Russian Academy of Science, Moscow, Russia
[5]Laboratory for Ocean and Climate Dynamics, Pilot National Laboratory for Marine Science and Technology, Qingdao, China

**Correspondence:** Qiang Wang (Qiang.Wang@awi.de)

**Abstract.** In this paper we studied the impact of winds on Arctic sea ice through the ocean's memory by using numerical simulations. We found that the changes in halosteric height induced by wind perturbations can significantly affect the Arctic sea ice drift, thickness, concentration and deformation rates regionally even years after the wind perturbations. Changes in the Arctic liquid freshwater content thus in halosteric height can cause changes in the sea surface height and surface geostrophic
currents, which further enforce a lasting and strong impact on sea ice. Both the changes in sea surface height gradient force (due to changes in sea surface height) and ice-ocean stress (due to changes in surface geostrophic currents) are found to be important in determining the overall ocean effects. The revealed ocean effects are mainly associated with changes in sea ice dynamics, not thermodynamics. Depending on the preceding atmospheric mode driving the ocean, the ocean's memory of the wind forcing can lead to changes in Arctic sea ice characteristics with very different spatial patterns. We obtained these spatial
patterns associated with Arctic Oscillation, Arctic Dipole Anomaly and Beaufort High modes through dedicated numerical simulations. The dynamical impact of the ocean has strong seasonal variations, stronger in summer and weaker in winter and spring. It implies that declining trends of Arctic sea ice will very possibly allow a stronger ocean impact on the sea ice in a warming climate.

## 1 Introduction

Arctic sea ice has undergone significant changes over the period of satellite observations. Not only the Arctic sea ice coverage but also the Arctic sea ice thickness has declined dramatically (Stroeve et al., 2012; Laxon et al., 2013; Kwok, 2018; Comiso et al., 2017), with potential impacts on the Northern Hemisphere weather and climate (e.g., Vihma, 2014; Wunderling et al., 2020). Contemporarily, sea ice drift in the Arctic Ocean was observed to speed up (Rampal et al., 2009; Spreen et al., 2011; Kwok et al., 2013; Petty et al., 2016), whereas the transported sea ice in the Transpolar Drift and the sea ice volume export
through Fram Strait (see Figure 1 for Arctic geographical features) have been decreasing due to sea ice thinning (Krumpen et al., 2019; Spreen et al., 2020; Wang et al., 2021a).

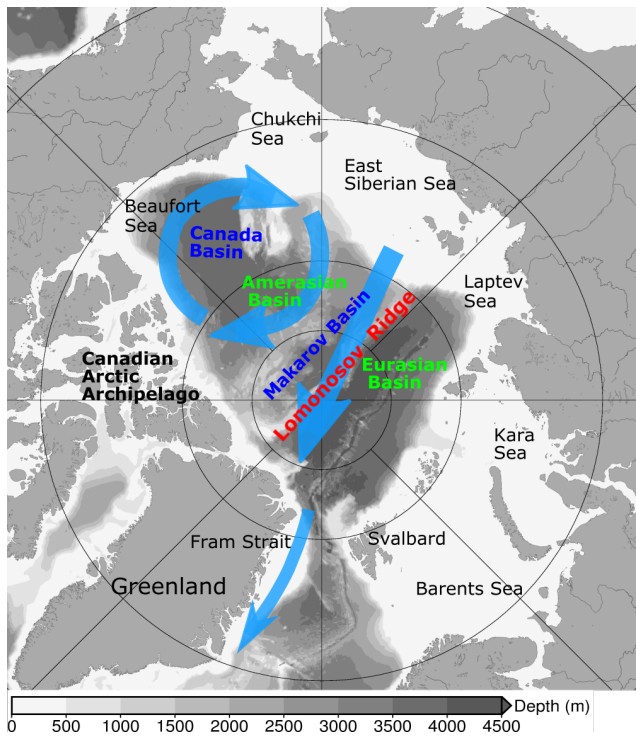

**Figure 1.** Arctic geographic features and schematic of Arctic sea ice circulation (the anticyclonic Beaufort Gyre circulation over the Canada Basin, the Transpolar Drift across the central Arctic, and the sea ice export through the Fram Strait; shown as blue arrows). The background gray color shows bottom bathymetry.

The decline in Arctic sea ice thickness and extent is accompanied by pronounced variability on different time scales, with contribution from both dynamic and thermodynamic processes (Serreze and Meier, 2019). Although the variability of Arctic sea ice area is largely determined by atmospheric temperature fluctuations (Olonscheck et al., 2019), wind forcing also plays an important role. For example, wind variation associated with the Arctic Dipole Anomaly (DA) can significantly influence the Transpolar Drift, thus affecting Fram Strait sea ice export and Arctic summer sea ice extent (Wu et al., 2006; Wang et al., 2009; Kwok et al., 2013; Platov et al., 2020).

The Arctic liquid freshwater content varies on a quasi-decadal time scale as a memory of wind forcing (Proshutinsky et al., 2015; Johnson et al., 2018). Over the last two decades, the Arctic Ocean has accumulated an unprecedented amount of liquid freshwater in the Amerasian Basin (McPhee et al., 2009; Giles et al., 2012; Rabe et al., 2014; Proshutinsky et al., 2019) due to the combination of a dominant anticyclonic wind regime, enhanced momentum transfer resulting from sea ice decline and a freshening of source waters (Krishfield et al., 2014; Davis et al., 2014; Wang et al., 2019c). The associated increase in the halosteric height led to a sea surface height doming in the Beaufort Gyre, which intensified the anticyclonic surface geostrophic current (McPhee, 2013; Armitage et al., 2016, 2017). Moreover, increasingly stronger currents in the upper ocean of the eastern

Eurasian Basin were also observed over the past decades (Polyakov et al., 2020), which can be explained by a drop in freshwater content in this basin associated with sea ice decline (Wang et al., 2019c).

     The ocean surface current and sea ice drift are strongly coupled through the ice-ocean stress (e.g., Tsamados et al., 2014; Heorton et al., 2019). Sea ice has strong internal stress seasonally, so its state can significantly influence the ocean circulation through changing the ice-ocean stress (Martin et al., 2016). In the Beaufort Gyre region, sea ice can limit the spinup of the ocean

circulation when ocean surface geostrophic velocity exceeds sea ice drift (Dewey et al., 2018; Zhong et al., 2018; Meneghello et al., 2018; Wang et al., 2019a). As Arctic sea ice declines, the response of sea ice drift to wind variability intensifies, which can thus strengthen the variability of the Arctic sea surface height and surface geostrophic currents (Wang, 2021). A significant part of the sea ice motion averaged over several months can be due to ocean surface currents (Thorndike and Colony, 1982). In particular, it was found that the spinup of the ocean can accelerate sea ice drift in the Beaufort Gyre region, especially in warm

seasons when sea ice internal stress is low (McPhee, 2013; Kwok and Morison, 2017; Wang et al., 2019a), which may cause sea ice export from the gyre (McPhee, 2013).

     With continuing climate change there is an increasing need to predict Arctic sea ice conditions on a variety of temporal and spatial scales (Jung et al., 2016; Serreze and Meier, 2019). An improved understanding of possible impacts of changes in the ocean on sea ice could provide useful information for sea ice predictions. In terms of oceanic thermal forcing, previous studies

suggest that ocean heat from sub-Arctic seas can accelerate Arctic sea ice decline in a warming climate (e.g., Polyakov et al., 2017; Årthun et al., 2019; Shu et al., 2021). The impacts of oceanic dynamic forcing on Arctic sea ice on seasonal to multiyear scales, namely through changes in sea surface height and surface geostrophic currents, still need a better understanding. Such understanding would also be helpful for interpreting observed regional changes in Arctic sea ice in terms of natural variability versus climate change signals, because sea surface height in the Arctic Ocean varies on interannual to decadal scales (Koldunov

et al., 2014; Armitage et al., 2016; Xiao et al., 2020).

     In this paper we will use high resolution numerical simulations with a global sea ice - ocean model to investigate the dynamical impact of the ocean on Arctic sea ice drift, thickness and concentration. We carried out both control and sensitivity experiments, which are the same except for the initial ocean states that have different sea surface height and surface geostrophic currents in the Arctic Ocean. The initial ocean states of the sensitivity simulations were obtained by applying wind perturbations

representing different Arctic atmospheric circulation modes beforehand. By comparing the sensitivity simulations with the control simulation, we identified the impact of ocean states and different preceding wind forcings on sea ice.

     The method and model setups are described in Section 2, and results are presented in Section 3. Discussion and conclusions are provided in Section 4 and 5, respectively.

## 2    Method and model setups

We used the global Finite Element Sea-ice Ocean Model (FESOM 1.4, Wang et al., 2014) in this paper. Both the ocean and sea ice components of FESOM 1.4 work with unstructured triangular meshes, allowing for multi-resolution global simulations (Danilov et al., 2004; Wang et al., 2008; Danilov et al., 2015). The elastic-viscous-plastic (EVP, Hunke and Dukowicz,

1997) sea ice rheology with improved convergence (Danilov et al., 2015) and the Parkinson and Washington (1979) sea ice thermodynamics are used in the model version employed here. The K-profile parameterization scheme (Large et al., 1994) and Smagorinsky viscosity (Smagorinsky, 1963) in a biharmonic form are used for ocean diapycnal mixing and horizontal viscosity, respectively. Eddy diffusivity varies with local horizontal resolution as suggested by Wang et al. (2014). The model has been widely used in studying Arctic sea ice and ocean (e.g., Wang et al., 2016a; Wekerle et al., 2017a, b; Wang et al., 2018, 2019b, 2020, 2021a) and evaluated in these studies.

We employed the multi-resolution model grid that has been used in Wang (2021). The horizontal resolution is $1^\circ$ in most parts of the global ocean. It is refined to 24 km north of $45^\circ$N and further refined to 4.5 km inside the Arctic Ocean. The grid has 47 z-levels in total. The vertical spacing is 10 m in the upper 100 m and gradually coarsened downward. The simulations were done with a time step of 12 minutes.

A control simulation was performed from 1958 to 2019 using atmospheric forcing from the JRA55-do data set (Tsujino et al., 2018). This forcing has a spatial resolution of $0.55^\circ$ and a temporal resolution of 3 hours. It was shown that many different community sea ice-ocean models using this data set can reasonably reproduce the observed changes in the ocean and sea ice (Tsujino et al., 2020). The control simulation was initialized from the PHC 3 climatology (Steele et al., 2001) and climatological sea ice derived from a previous simulation (that is, December sea ice averaged over 1970 - 1990 obtained from a simulation with the same model configuration). The trend and interannual variability in Arctic sea ice volume and summer sea ice extent over the last four decades are reasonably simulated in the model in comparison to observations and reanalysis, and the simulated sea ice concentration, thickness and drift also compare well with satellite observations (Figures S1 - S4, Schweiger et al., 2011; Lavergne et al., 2010, 2019; Fetterer et al., 2017; Hendricks and Ricker, 2019). This model configuration can also well reproduce the trend and variability of the Arctic sea surface height observed by satellites and tide gauges (Xiao et al., 2020) and the recent changes in Arctic freshwater content (Wang, 2021).

To prepare sensitivity simulations, we first carried out six simulations with wind perturbations added to wind forcing for the calculation of wind stress. They were performed for six years from 2010 to 2015 starting from the control run results. Using specifically designed wind perturbations representing Arctic major atmospheric modes (see the description below), the Arctic liquid freshwater content and thus sea surface height and surface geostrophic currents can be accordingly perturbed (Wang, 2021). The sensitivity simulations were then performed for four years from 2016 to 2019 starting from the perturbed ocean states. The initial sea ice conditions of the sensitivity runs were taken from the control run, and the wind perturbations were turned off. That is, the four-year-long sensitivity simulations are the same as the control run except that different initial ocean states are used. The difference of the model results between the sensitivity experiments and the control run can reveal the impacts of prior wind perturbations on sea ice through the ocean's memory.

The wind perturbations were designed as described below. The first two empirical orthogonal functions (EOF) of deseasonalized (mean seasonal cycle removed) monthly sea level pressure (SLP) north of $70^\circ$N were calculated over the period of 1980 - 2019 using the JRA55-do data set (Figures 2a and 2d). EOF1 resembles the negative, anticyclonic phase of the Arctic Oscillation (AO, Thompson and Wallace, 1998) and explains 68% of the SLP variability. EOF2 represents the DA mode (Wu et al., 2006) and explains 13% of the SLP variability. The negative value of EOF2 shown in Figure 2d depicts the negative phase

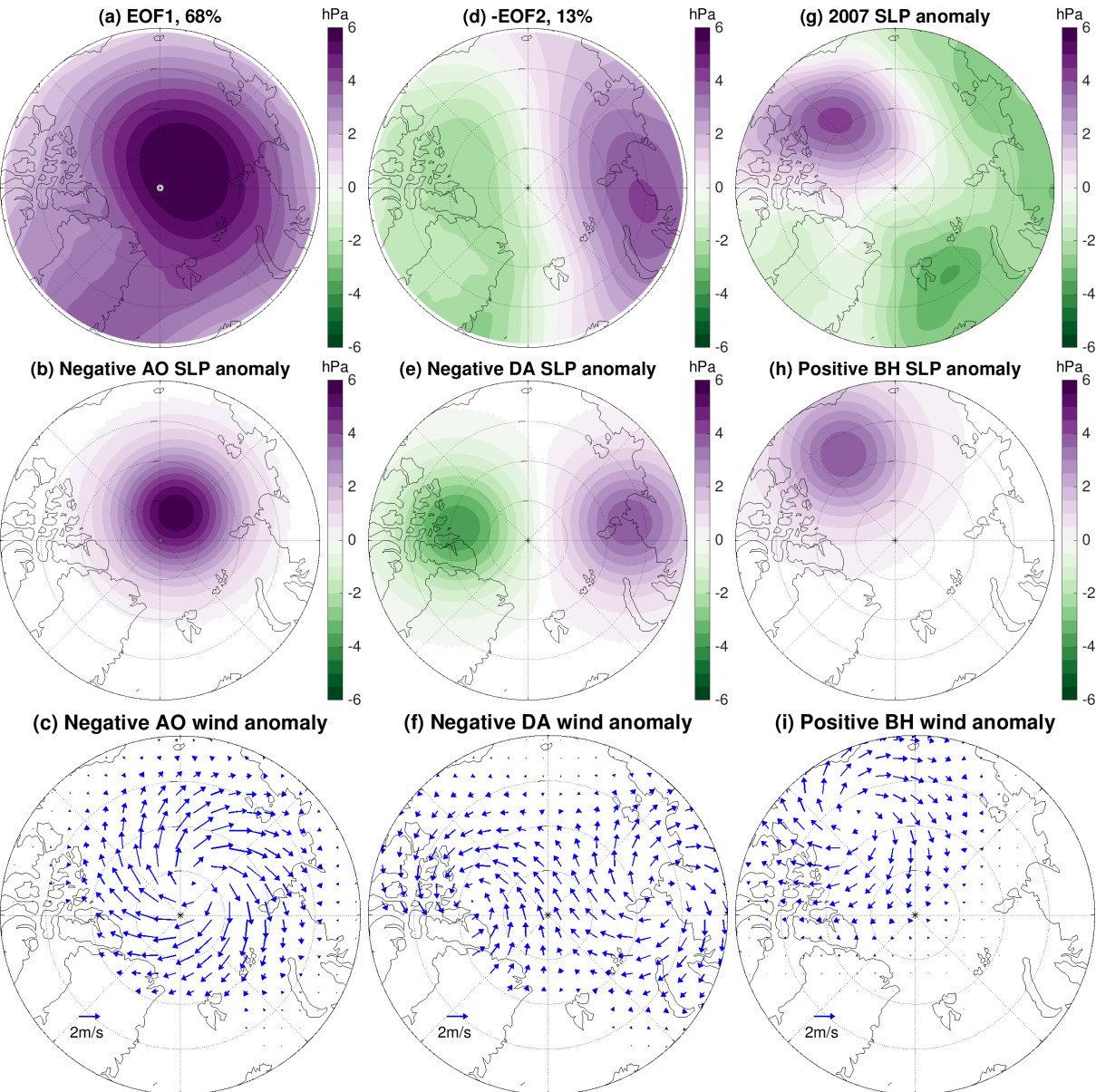

**Figure 2.** (a) The first empirical orthogonal function (EOF) of sea level pressure (SLP) for the period 1980-2019. It resembles the negative phase of the Arctic Oscillation (AO). (b) Idealized SLP anomaly representing negative AO and (c) the associated wind anomaly. (d) The second EOF of SLP. The negative EOF2 is shown, which represents the negative phase of the Arctic Dipole Anomaly (DA). (e) Idealized SLP anomaly representing negative DA and (f) the associated wind anomaly. (g) 2007 SLP anomaly relative to the mean over 1980-2019, which shows a strongly positive Beaufort High (BH) SLP anomaly. (h) Idealized SLP anomaly representing the positive BH phase and (i) the associated wind anomaly. The magnitudes of the idealized SLP anomalies are 6 hPa in the AO case and 4 hPa in the other cases.

of the DA. On average, the Beaufort High (BH) SLP was higher than normal in the early 21st century and caused a dramatic increase of liquid freshwater in the Beaufort Gyre region (e.g., McPhee et al., 2009; Proshutinsky et al., 2019). In particular,

the SLP over the Beaufort Gyre was in a strongly positive phase in 2007 (Figure 2g). For generating perturbed ocean states for the sensitivity experiments, the three atmospheric modes mentioned above (AO, DA, BH) were considered. Wind anomalies associated with three idealized SLP anomalies representing these modes were used: AO (Figure 2b,c), DA (Figure 2e,f) and BH (Figure 2h,i). The BH anomalies were adopted from Marshall et al. (2017). Both the negative and positive phases of these atmospheric modes were used, so we obtained six perturbed ocean states and performed six sensitivity experiments.

In order to disentangle the role of the dynamical forcing imposed by the ocean on sea ice, that is, ocean-ice stress and pressure gradient force due to sea surface tilt, we carried out a few additional experiments. They are the same as the sensitivity experiments described above, except that the sea surface height in the pressure gradient force term of the sea ice momentum equation is replaced with that saved from the control simulation (see details in Section 3.3). One more simulation was performed to compare the dynamic impact of the ocean on sea ice with the impact of the initial sea ice state. It is the same as the sensitivity

simulation with prior negative AO forcing described above, but with a different initial sea ice state (see details in Section 4.1).

## 3   Results

### 3.1   Perturbed ocean state

By applying the wind perturbations, the magnitudes and spatial patterns of Arctic halosteric height and sea surface height (the dynamical sea level simulated by the model) were changed, as shown by their anomalies relative to the control run in

the last (sixth) year of the wind-perturbation simulations (Figures S5 and S6) and in the sensitivity simulations after the wind perturbations were switched off (Figures 3 and 4). The changes in sea surface height can be explained by the changes in halosteric height, which are associated with the changes in liquid freshwater content (Giles et al., 2012; Armitage et al., 2016; Wang, 2021). The main dynamical processes changing Arctic freshwater content and halosteric height under wind perturbations are Ekman transport of freshwater, although induced changes in sea ice thermodynamics also have certain contributions (Wang,

125  2021).

Different atmospheric modes can lead to changes in the halosteric height and sea surface height with very different spatial patterns (Figures S3,S4 and Figures 3,4). With the negative AO perturbation, the sea surface height increases in the central Arctic including most parts of the Eurasian and Makarov basins, while it drops in the surrounding area (Figure S6a and Figure 4a). With the positive AO perturbation, opposite changes are found (Figure S6b and Figure 4b). With the DA forcing,

the magnitudes of the halosteric height and sea surface height anomalies are smaller than with the other perturbations. The negative DA perturbation increases the sea surface height in the eastern Eurasian Basin and along the southwestern periphery of the Amerasian Basin (Figure S6c and Figure 4c), and the changes induced by the positive DA perturbation are roughly opposite (Figure S6d and Figure 4d). With the BH forcing, the sea surface height changes oppositely between the Canada Basin and Eurasian Basin (Figure S6e,f and Figure 4e,f). A positive BH anomaly increases sea surface height in the Canada

Basin and reduces it in the Eurasian Basin.

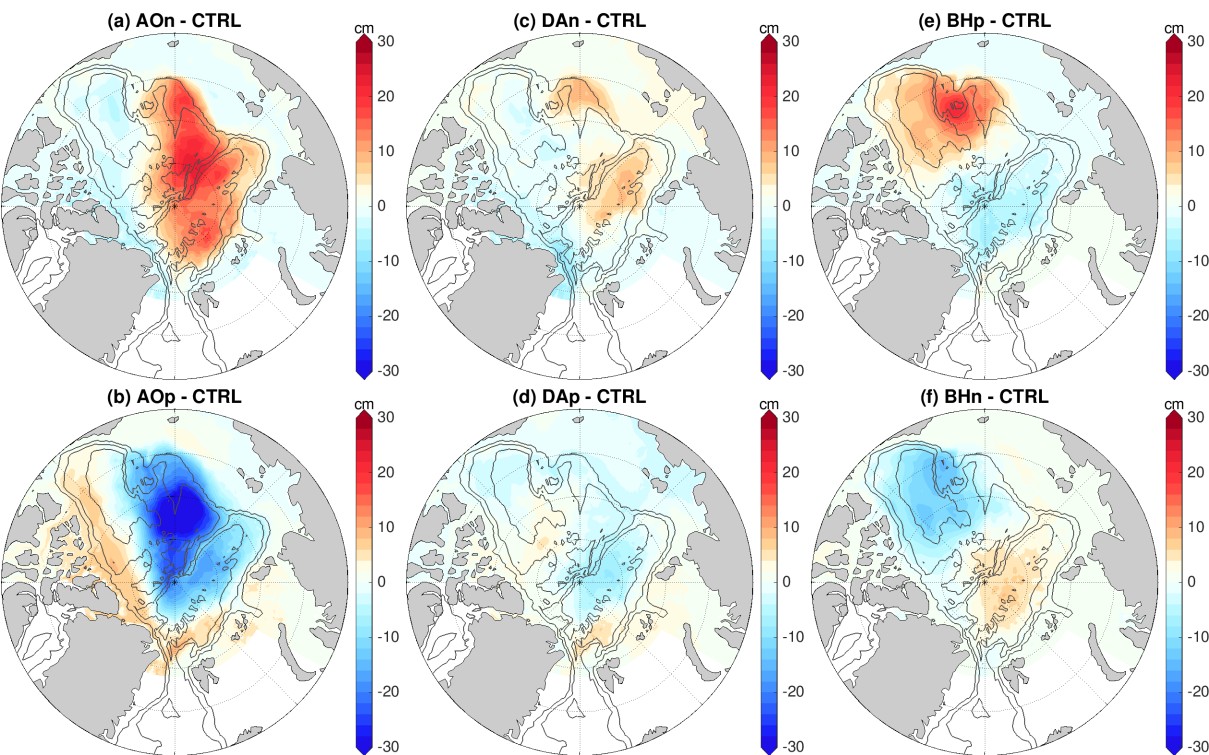

**Figure 3.** Anomaly of halosteric height (cm) relative to the control run averaged over the four model years of the sensitivity simulations in which wind perturbations were switched off: Experiments with an initial ocean spun up with (a) negative phase of Arctic Oscillation (AO) forcing, (b) positive phase of AO forcing, (c) negative phase of Dipole Anomaly (DA) forcing, (d) positive phase of DA forcing, (e) positive phase of Beaufort High (BH) forcing, and (f) negative phase of BH forcing. The halosteric height is referenced to 400 m depth or the ocean bottom if it is shallower than 400 m. As the wind perturbations were switched off in the simulations, the anomalies indicate the ocean memory of the wind perturbations applied beforehand. The gray contour lines indicate the 500-, 2000- and 3500-m isobaths.

Because the wind perturbations were turned off during the sensitivity simulations, the magnitudes of the anomalies of halosteric height and sea surface height relative to the control run decreased in most of the areas in the sensitivity simulations, but the spatial patterns of the anomalies largely remained (compare Figures 3 and 4 with Figures S5 and S6). Therefore, the ocean kept a memory of the wind perturbations applied before.

The time series of sea surface height anomalies averaged in specific regions possessing typical ocean changes induced by the wind perturbations are shown in Figure 5a-c. They depict the evolution of the sea surface height during the perturbation simulations (the first 6 years) and afterwards in the sensitivity simulations (the last 4 years). Consistent with the spatial patterns of the anomalies (Figure 4), the changes in area-mean sea surface height are quasi-symmetric between the negative and positive perturbations for different forcing cases (Figure 5a-c). Differences in the magnitudes of the sea surface height anomalies
between opposite forcing phases are present regionally. For example, the magnitudes of the sea surface height anomalies are higher in the positive than in the negative AO forcing cases (Figure 4a,b). The magnitudes of the sea surface height anomalies

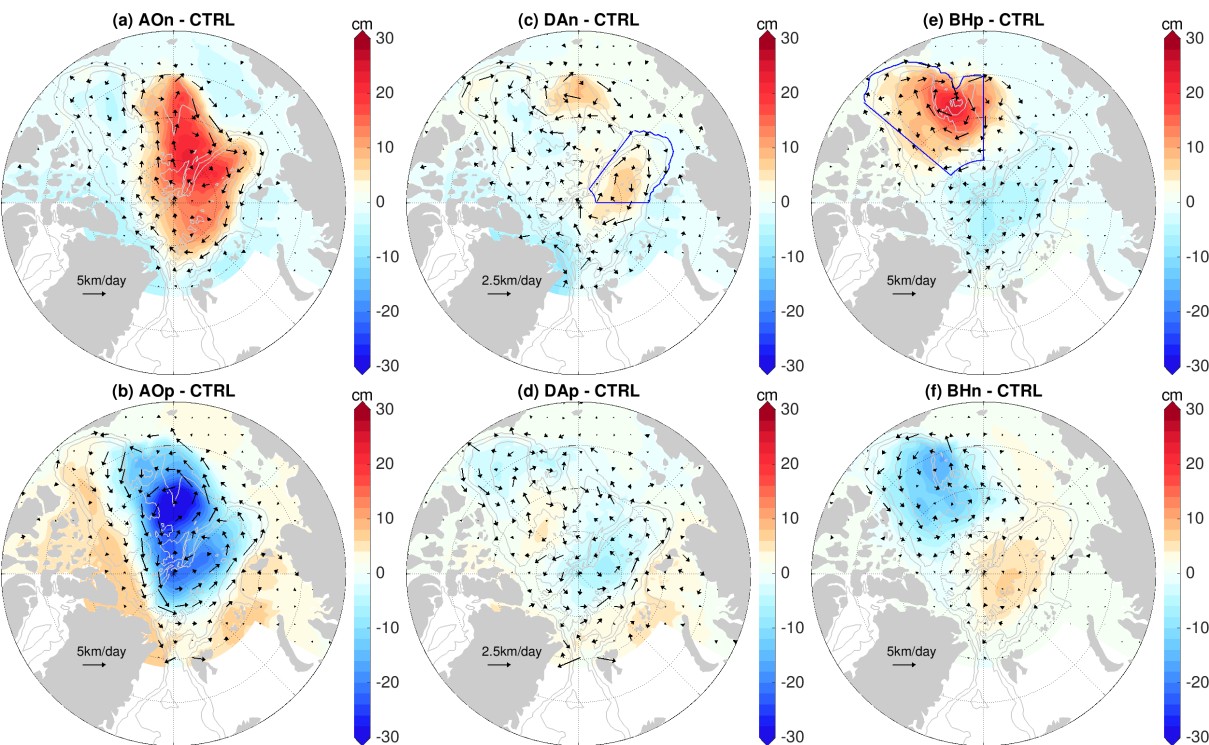

**Figure 4.** Anomaly of simulated sea surface height (SSH, patch color) and surface geostrophic current (arrows) relative to the control run averaged over the four model years of the sensitivity simulations in which wind perturbations were switched off: Experiments with an initial ocean spun up with (a) negative phase of Arctic Oscillation (AO) forcing, (b) positive phase of AO forcing, (c) negative phase of Dipole Anomaly (DA) forcing, (d) positive phase of DA forcing, (e) positive phase of Beaufort High (BH) forcing, and (f) negative phase of BH forcing. Note that the scaling for velocity arrows in the DA cases is different from other cases. The SSH anomalies can be explained by the halosteric height anomalies shown in Figure 3. The gray contour lines indicate the 500-, 2000- and 3500-m isobaths. The blue lines in (c) and (e) indicate the eastern Eurasian Basin and the Canada Basin, respectively, which are used in Figure 5.

decrease with time in the sensitivity simulations, but they are still relatively large at the end of the sensitivity simulations, especially in the cases with initial ocean obtained with AO and BH forcing (Figure 5a-c). The sea surface height anomalies have negligible seasonal variation in all the sensitivity simulations (Figure 5d).

## 3.2   Impact on sea ice

As the only difference between the settings of the sensitivity simulations and the control run is the ocean initial conditions obtained by applying wind perturbations beforehand, the anomalies of the sensitivity simulations relative to the control run can be attributed to the impact of prior wind perturbations through the ocean's memory.

We found that significant changes are induced in sea ice drift by the perturbed ocean (Figure 6). The sea ice drift anomalies are largely aligned and scaled with the anomalies of surface geostrophic currents. In the case of prior negative AO perturbation,

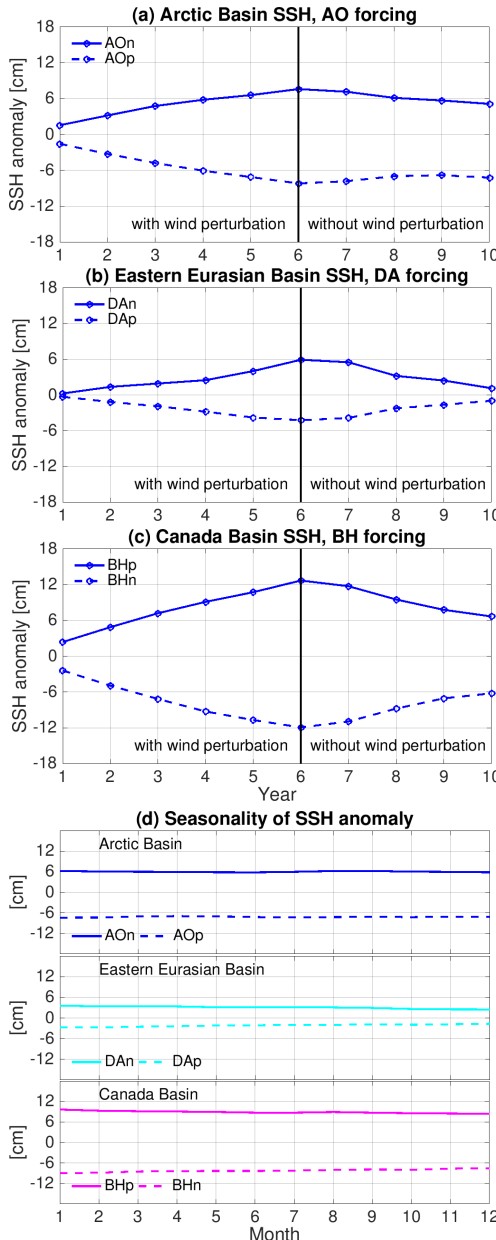

**Figure 5.** (a) Anomaly of annual mean sea surface height (SSH, cm) averaged in the Arctic Basin (the area where bottom bathymetry is deeper than 500 m) in the simulations with Arctic Oscillation (AO) perturbations. The anomalies are referenced to the control run. (b) The same as (a), but for the mean in the eastern Eurasian Basin (indicated by blue lines in Figure 4c) in the simulations with Dipole Anomaly(DA) perturbations. (c) The same as (a), but for the mean in the Canada Basin (indicated by blue lines in Figure 4e) in the simulations with Beaufort High (BH) perturbations. (d) The mean seasonal cycle of the SSH anomaly (the difference between the sensitivity runs and the control run) in the last four years, which indicates that the seasonal variation of the SSH anomaly is negligible. Note that *the SSH in each individual simulation* does have clear seasonal variability (not shown).

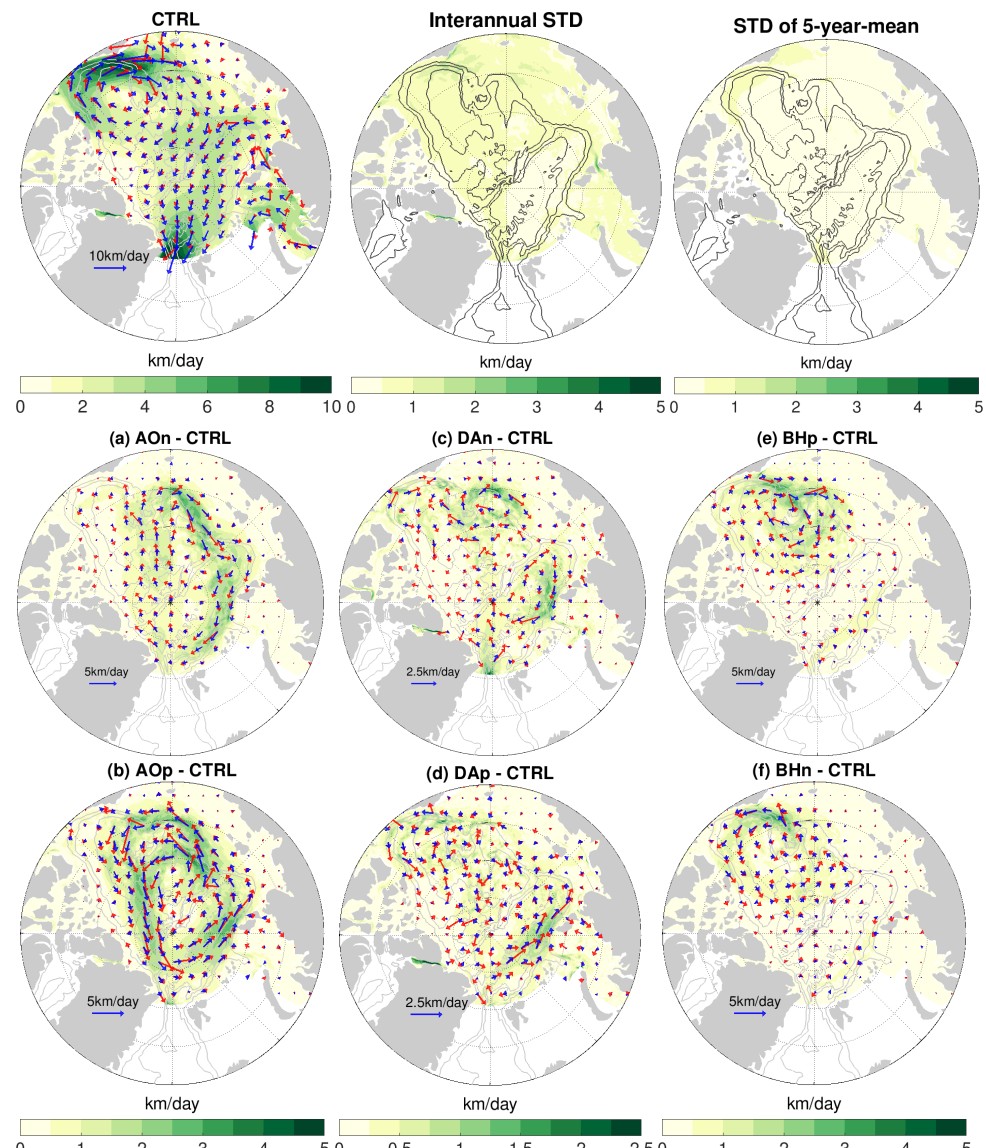

**Figure 6.** (a)-(f) Anomaly of sea ice drift (blue arrows) and ocean surface geostrophic current (red arrows) averaged over the four model years of the sensitivity simulations in which wind perturbations were switched off: Experiments with an initial ocean spun up with (a) negative phase of Arctic Oscillation (AO) forcing, (b) positive phase of AO forcing, (c) negative phase of Dipole Anomaly (DA) forcing, (d) positive phase of DA forcing, (e) positive phase of Beaufort High (BH) forcing, and (f) negative phase of BH forcing. The magnitude of the ice drift anomaly is also shown with color. The anomalies are referenced to the control run. The mean sea ice drift in this period (left), the standard deviation (STD) of sea ice drift speed on the interannual time scale in the 2010s (middle), and the STD of the pentadal mean in the period 1980-2019 (right) from the control run are shown on top of the figure for reference. Note that the scaling for velocity arrows and the range for color are different in different panels, but in each panel the same scaling is used for sea ice drift and geostrophic current. The gray contour lines indicate the 500-, 2000- and 3500-m isobaths.

the sea ice drift anomalies are anticyclonic around the Arctic basin with magnitudes of 1.5-3 km/day (Figure 6a). In the positive AO case, the sea ice drift anomalies are cyclonic (Figure 6b). The magnitudes of the drift anomalies are larger in the case of prior positive AO perturbation, as the sea surface height and surface geostrophic currents imply (Figure 4a,b). In the cases of prior DA perturbations, the magnitudes of sea ice drift anomalies are smaller than in other cases, and the anomalies are also less regular in space (Figure 6c,d), as expected from the sea surface height anomalies (Figure 4c,d). In the Eurasian Basin, the sea ice drift anomalies are anticyclonic (cyclonic) as the fingerprints of the prior negative (positive) DA forcing. In the cases of prior BH perturbations, the largest sea ice drift anomalies are found in the Amerasian Basin (Figure 6e,f). The anomalies are anticyclonic (cyclonic) around the Beaufort Gyre in the case of prior positive (negative) BH forcing with magnitudes of about 1-2 km/day.

The significance of the impact on sea ice drift can be judged by comparing the anomalies with the mean values and the variability of sea ice drift in the control run (Figure 6, top row). In the Beaufort Gyre region, the sea ice drift anomalies induced by the ocean states from prior BH perturbations amount to about 25% of the mean sea ice drift in the control run. The sea ice drift anomalies in the case of prior AO forcing can even have magnitudes locally similar to those of the mean sea ice drift, although their directions are mostly not the same. Furthermore, the sea ice drift anomalies induced by the perturbed ocean are regionally much larger than the standard deviation of both annual and 5-year-mean sea ice drift.

The perturbed ocean also causes profound changes in sea ice thickness (Figure 7). The induced sea ice thickness anomalies have different spatial patterns and magnitudes in different cases. In the case of prior negative AO perturbation, we found positive sea ice thickness anomalies from north of the Canadian Arctic Archipelago to the eastern Canada Basin with magnitudes up to 20-30 cm, and negative anomalies in the western Eurasian Basin and central Arctic with magnitudes up to 10-15 cm (Figure 7a). In the case of prior positive AO perturbation, the sea ice thickness anomalies are opposite, with larger magnitudes (Figure 7b) in accordance with a stronger ocean perturbation (Figure 4b). The ocean perturbed with DA forcing causes opposite changes in sea ice thickness between in the Eurasian Basin and north of the Canadian Arctic Archipelago, with magnitudes up to about 10 cm (Figure 7c,d), which are less pronounced than in other forcing cases. In the case of negative DA perturbation, the sea ice thickness anomalies are negative in the Eurasian Basin and positive north of the Canadian Arctic Archipelago. In the cases of prior BH perturbations, largest sea ice thickness anomalies are found in the Canada Basin with magnitudes up to 15-20 cm and in the western Eurasian Basin with magnitudes up to about 10-15 cm (Figure 7e,f). With the ocean perturbed with the positive BH perturbation, the sea ice thickness anomaly is negative in the Canada Basin and positive in the western Eurasian.

The sea ice thickness anomalies induced by the perturbed ocean regionally can reach about 10% of the mean sea ice thickness in the control run (Figure 7, top-left panel), for example, north of the Canadian Arctic Archipelago in the cases of prior AO perturbations and in the Canada Basin in the cases of prior BH perturbations. In these places, the magnitudes of the anomalies in the corresponding AO and BH forcing cases are comparable with the standard deviation of 5-year-mean sea ice thickness of the control run, and slightly smaller than the standard deviation of annual mean sea ice thickness (Figure 7, middle and right panels in the top row). Therefore, these sea ice thickness anomalies are significant considering that they persist for years following the ocean anomalies. The sea ice thickness anomalies in the cases of prior DA forcing are significant only in very small coastal areas adjacent to the Canadian Arctic Archipelago and Greenland.

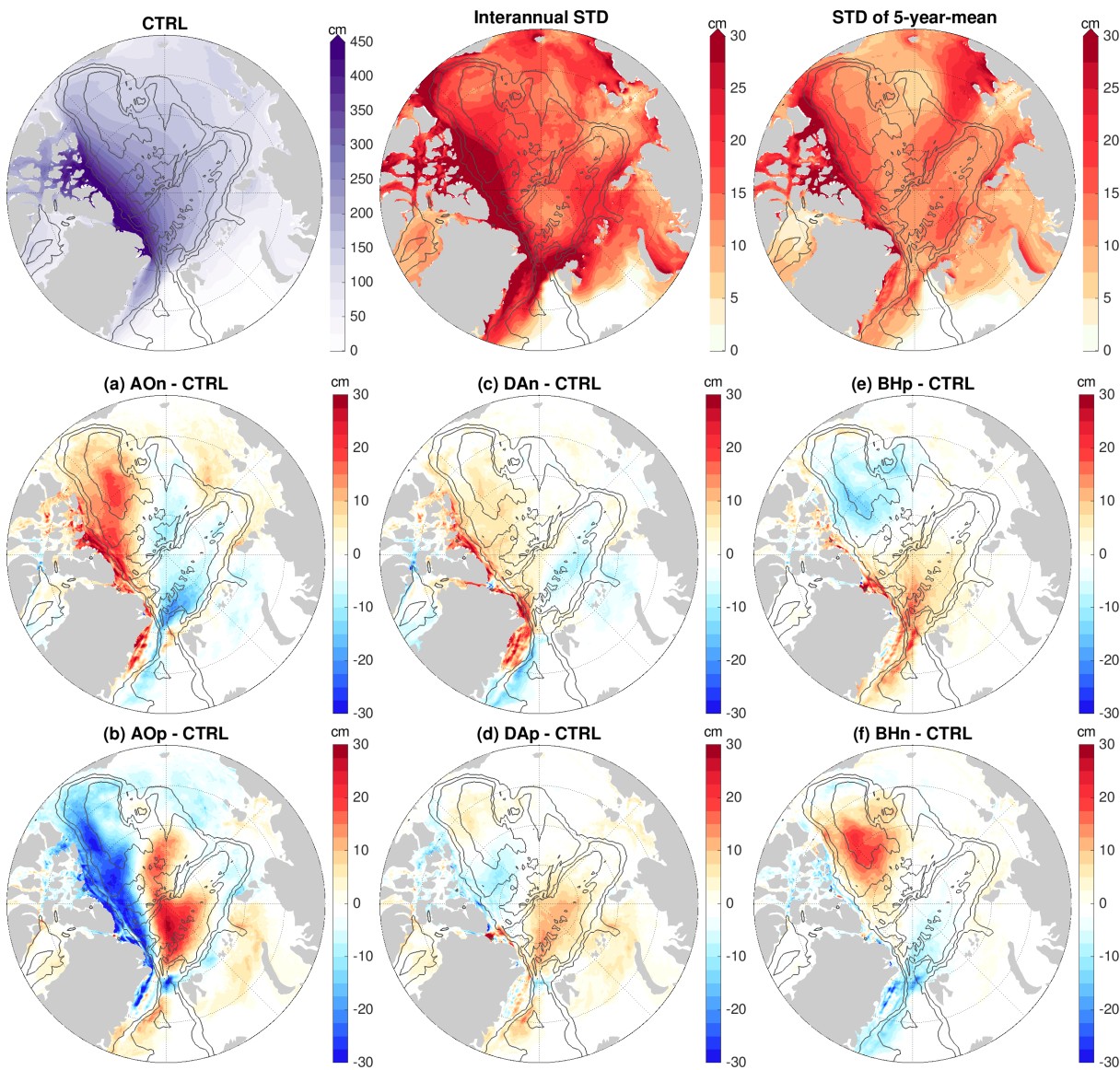

**Figure 7.** (a)-(f) Anomaly of sea ice thickness averaged over the four model years of the sensitivity simulations in which wind perturbations were switched off: Experiments with an initial ocean spun up with (a) negative phase of Arctic Oscillation (AO) forcing, (b) positive phase of AO forcing, (c) negative phase of Dipole Anomaly (DA) forcing, (d) positive phase of DA forcing, (e) positive phase of Beaufort High (BH) forcing, and (f) negative phase of BH forcing. The anomalies are referenced to the control run result. The mean sea ice thickness in this period (left), the standard deviation (STD) of sea ice thickness on the interannual time scale in the 2010s (middle), and the STD of the pentadal mean in the period 1980-2019 (right) from the control run are shown on top of the figure for reference. The black contour lines indicate the 500-, 2000- and 3500-m isobaths.

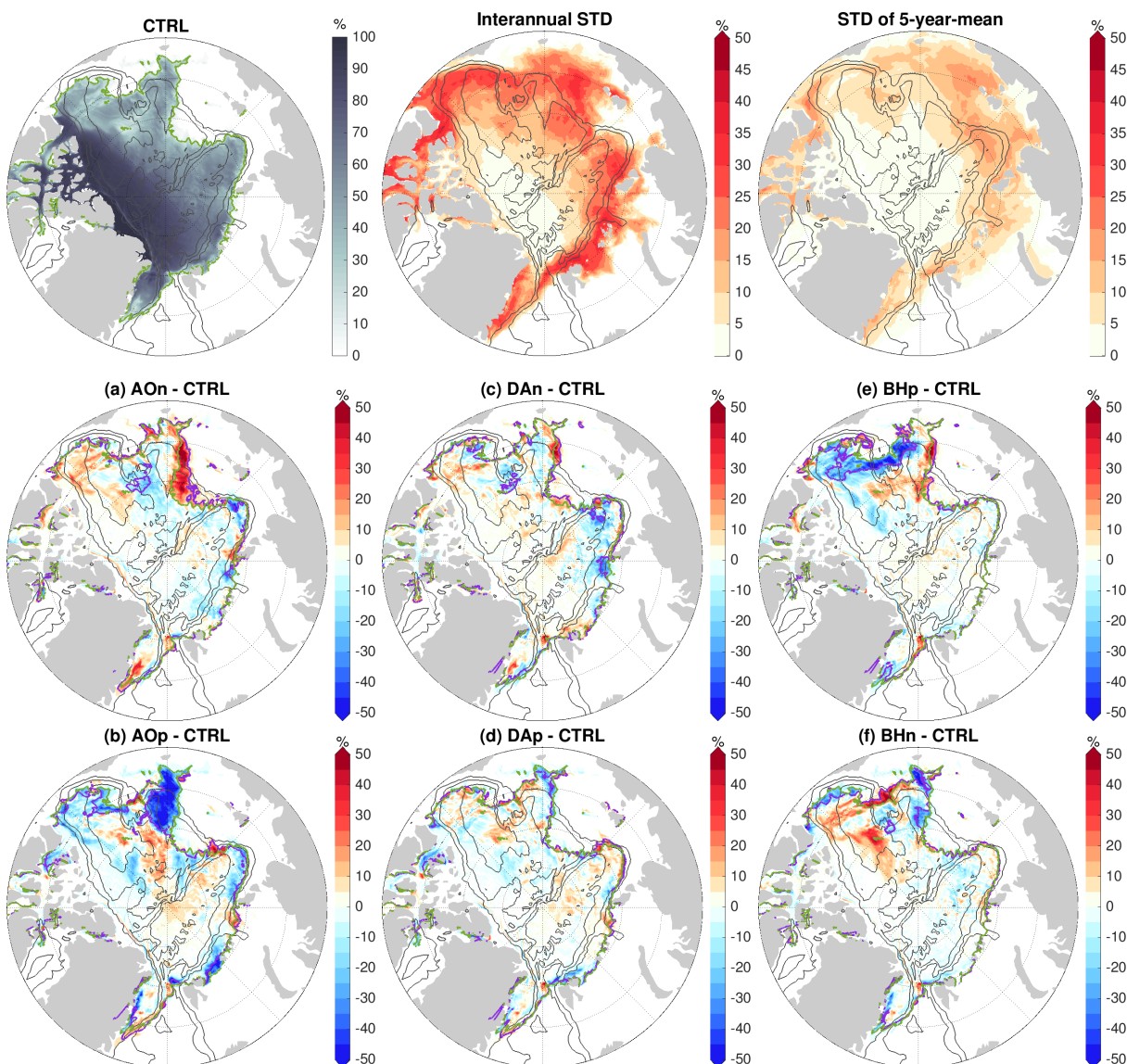

**Figure 8.** (a)-(f) September sea ice concentration anomaly in the first year (2016) of the sensitivity simulations in which wind perturbations were switched off: Experiments with an initial ocean spun up with (a) negative phase of Arctic Oscillation (AO) forcing, (b) positive phase of AO forcing, (c) negative phase of Dipole Anomaly (DA) forcing, (d) positive phase of DA forcing, (e) positive phase of Beaufort High (BH) forcing, and (f) negative phase of BH forcing. The anomaly is referenced to the control run. The September sea ice concentration in this year (left), the standard deviation (STD) of September sea ice concentration on the interannual time scale in the 2010s (middle), and the STD of pentadal mean in the period 1980-2019 (right) from the control run are shown on top of the figure for reference. The locations of sea ice edge (15% sea ice concentration) are indicated with green lines for the control run and violet lines for the sensitivity runs. The black contour lines indicate the 500-, 2000- and 3500-m isobaths.

We found that the influence of the perturbed ocean states on sea ice concentration is only pronounced in summer. The anomalies of September sea ice concentration in the first year of the sensitivity simulations relative to the control run are shown in Figure 8. They indicate that the perturbed ocean can change sea ice concentration by more than 50% regionally and shift the location of sea ice edge in the southern and western Amerasian Basin, most strongly only in the cases of prior AO and BH forcing. In the case of prior negative AO perturbation, the sea ice edge in the western Amerasian Basin is located further to the west compared to the control run (Figure 8a), consistent with the anticyclonic sea ice drift anomalies in this region (Figure 6a). In the opposite case with prior positive AO perturbation, the sea ice edge retreats northeastward in the southwestern Amerasian Basin (Figure 8b), which is consistent with the cyclonic sea ice drift anomalies there (Figure 6b). The ocean perturbed by positive BH perturbation causes sea ice edge to retreat northward in the southwestern Amerasian Basin and to slightly expand westward in the western Amerasian Basin (Figure 8e), consistent with the anticyclonic sea ice drift anomalies in the Beaufort Gyre (Figure 6e). Opposite impact is found in the case of prior negative BH perturbation, although the strength is weaker (Figure 8f). The sea ice concentration anomalies regionally can be larger than the magnitudes of the variability in the control run (Figure 8, top row), but they are confined to small areas close to ice edge. Because the impact on sea ice concentration is only significant in summer and close to sea ice edges, whose locations vary strongly in time, averaging the sea ice concentration anomalies over the four model years would mask the ocean impact. Therefore, in Figure 8 we showed the September sea ice concentration anomaly in one particular year.

The impact of the perturbed ocean on the Arctic total September sea ice extent is not large (figure not shown). The largest impact is in the cases of prior AO perturbations, with an increase of only about 3% for negative AO and a reduction of about 6% for positive AO, which are associated with the changes in the ice edge locations (Figure 8a,b). In the case of prior positive BH perturbation, the Arctic total sea ice extent is reduced by about 3%, while the change is negligible in the case of prior negative BH perturbation because the induced negative and positive sea ice concentration anomalies along the sea ice edge nearly compensate each other (Figure 8e,f). The impact on sea ice concentration and sea ice edge locations weakens when the ocean anomalies weaken, as depicted by the September sea ice concentration anomalies in the last year of the sensitivity simulations (Figure S7). In addition, the natural interannual variability of sea ice conditions could also influence the strength of the ocean effects. For example, the September sea ice concentration in the western Arctic is lower in the first model year of the sensitivity run (2016) than in the last year (2019). The less compact sea ice in 2016 potentially allows stronger impacts from the ocean.

The monthly root-mean-square (RMS) differences of sea ice drift, thickness and concentration between the sensitivity and control simulations are shown in Figure 9. The impact of the ocean on sea ice has strong seasonal variations. The impact is the largest in September and October, drop quickly in November and December, and stay at a relatively low level from January to May. The RMS difference of sea ice drift averaged over the Arctic Ocean is in the range of 0.5-1 km/day in winter and spring and 1.5-2 km/day in summer in different cases (Figure 9a), and the RMS difference of sea ice thickness is in the range of 7-15 cm in winter and spring and 15-30 cm in summer (Figure 9b). Compared with the seasonal Arctic mean sea ice drift between about 3 and 6 km/day and sea ice thickness between about 1 and 2 m, the RMS differences are notable in all seasons. The RMS sea ice concentration is very small (<3%) in winter, but it reaches the range of 5-10% in summer (Figure 9c). As shown above,

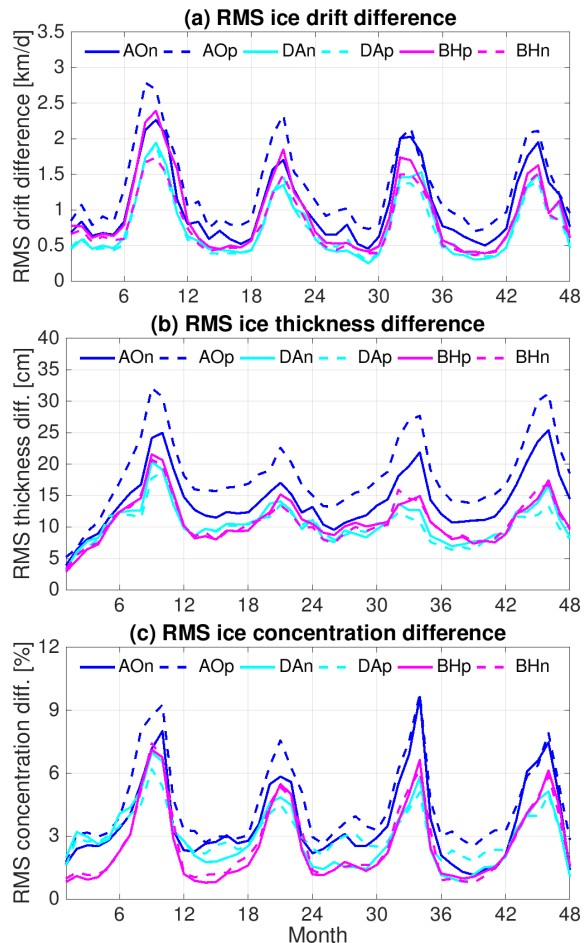

**Figure 9.** Monthly root-mean-square (RMS) difference of (a) sea ice drift speed, (b) thickness and (c) concentration between runs with perturbed initial ocean and the control run. The RMS difference is calculated where sea ice concentration is larger than 20% in both the simulations that are compared.

significant anomalies of summer sea ice concentration are mainly located close to the rim of the sea ice cover, with influence on sea ice edge locations regionally (Figure 8).

The seasonal variability of the anomaly in sea surface height is negligible (Figure 5d), so the seasonal variation of the impact is associated with the seasonal changes in sea ice compactness. Sea ice internal stress changes exponentially with sea ice concentration. At high sea ice concentration, it can partially mask the impact of the ocean (ocean-ice stress and ocean surface gradient) on sea ice momentum balance and sea ice drift. The impact of sea ice thermodynamics associated with the perturbed ocean is relatively small compared with the dynamical impact of the perturbed ocean (see the next section), so the seasonal variations of sea ice thickness and concentration anomalies are largely in phase with that of sea ice drift (Figure 9).

There are also interannual changes in the strength of the ocean impact (Figure 9), despite the fact that the ocean anomalies decrease with time without strong interannual variability (Figure 5a-c). Therefore, the interannual changes in sea ice states (such as sea ice concentration, thickness, edge location influenced by atmospheric forcings) can influence the significance of the ocean impact, as the seasonality of sea ice states does.

## 3.3 Attribution of the impact

Within the first month of the sensitivity simulations, sea ice drift has adjusted to the perturbed ocean states with RMS anomalies reaching typical wintertime levels (Figure 9a), while the RMS anomalies of sea ice thickness increased with time until reaching their first maximum in the first summer of the sensitivity simulations (Figure 9b). The anomalies of September sea ice thickness in the first year of the sensitivity simulations relative to the control run are shown in Figure 10a-c. For brevity, we only show results from one phase of each forcing case. These anomalies indicate the total effects of the perturbed ocean from January to September, because the initial sea ice states are the same in the sensitivity and control simulations. They have spatial patterns similar to the mean anomalies averaged over the four model years (Figure 7), but with magnitudes twice as large.

The differences of the changes in sea ice thickness from January to September in the first year due to thermodynamics (melting and freezing) between the sensitivity and control simulations are shown in Figure 10d-f. Thermodynamic sea ice changes can be attributed to different processes. For example, moving more sea ice to a warmer region could enhance sea ice melting. Furthermore, changes in sea ice thickness caused by sea ice drift can change heat conduction, thus sea ice heat budget and sea ice thermodynamics (e.g., Spall, 2019). However, the total contributions to the sea ice thickness anomalies from sea ice thermodynamic changes are rather small (cf. Figure 10a-c and Figure 10d-f). This indicates that contributions from sea ice dynamics (that is, redistribution of sea ice through advection of sea ice and sea ice convergence associated with changes in sea ice drift) are dominant. Indeed, the anomalies of the September sea ice thickness show opposite changes in different Arctic regions in all three forcing cases (Figure 10a-c), reflecting the fact of sea ice redistribution. For example, in the case of prior negative AO perturbation, the positive anomalies north of the Canadian Arctic Archipelago are compensated by negative anomalies in the western Eurasian Basin and the central Arctic. Accordingly, we found that the Arctic total sea ice volume is not significantly changed by the perturbed ocean. Our analysis further shows that the mean seasonal variations in sea ice thickness anomalies (smaller in winter and larger in summer, Figure 9b) can also be mainly attributed to the direct effect of sea ice dynamics (Figure S8).

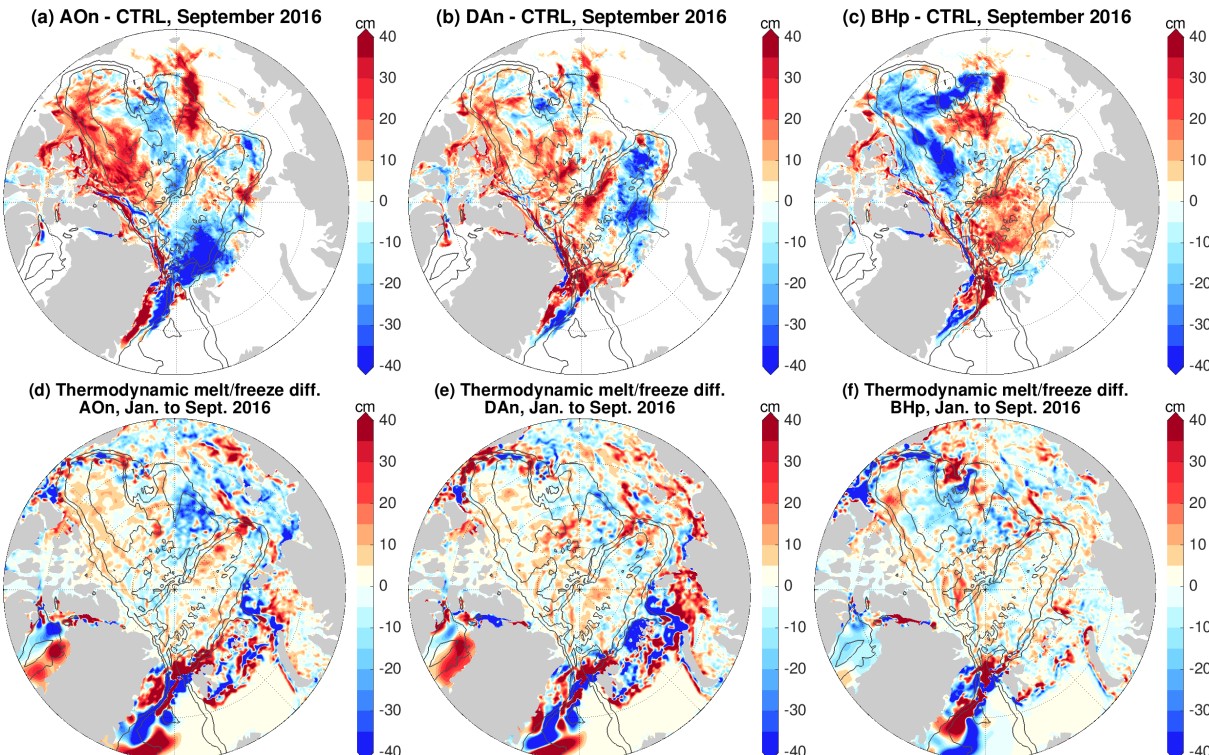

**Figure 10.** (a) Anomaly of sea ice thickness in September 2016 and (d) anomaly of the change in sea ice thickness from January to September in 2016 due to sea ice thermodynamics in the experiment with an initial ocean spun up with the negative phase of Arctic Oscillation (AO) perturbation beforehand. 2016 is the first model year of the sensitivity simulations. The anomalies are referenced to the control run. (b)(e) The same as (a)(d), but for the experiment with an initial ocean spun up with the negative phase of Dipole Anomaly (DA) perturbation; (c)(f) The same as (a)(d), but for the experiment with an initial ocean spun up with the positive phase of Beaufort High (BH) perturbation. The black contour lines indicate the 500-, 2000- and 3500-m isobaths.

260    The above analysis tells that the sea ice thickness anomalies are mainly induced by the direct sea ice dynamic effect in association with the changes in sea ice drift. In the following we will look into different ocean forcings that can influence sea ice drift and disentangle their impact on sea ice thickness. The ocean directly influences the sea ice drift through two forcing terms as shown by the sea ice momentum equation:

$$m(\partial_t + \mathbf{f} \times)\mathbf{u_i} = \alpha\tau + \alpha C_d \rho |\mathbf{u_o} - \mathbf{u_i}|(\mathbf{u_o} - \mathbf{u_i}) - mg\nabla\eta + \mathbf{F}, \tag{1}$$

265    where $m$ is mass per unit area, $f$ the Coriolis parameter, $\alpha$ the sea ice concentration, $\tau$ the wind stress, $C_d$ the ice-ocean drag coefficient, $\rho$ the water density, $\mathbf{u}_i$ the sea ice drift, $\mathbf{u}_o$ the ocean velocities, $g$ the gravity acceleration, $\eta$ the sea surface height, and $\mathbf{F}$ the sea ice internal stress divergence. The ocean influences the sea ice drift through the second (ice-ocean stress) and the third (sea surface height gradient force) terms on the right-hand-side of the equation.

The anomalies of sea ice drift are largely aligned with the anomalies of ocean surface geostrophic currents (Figure 6), which indicates that the perturbed ocean may influence sea ice drift mainly through ice-ocean stress. Does the sea surface height gradient force play an important role for sea ice drift and thickness then? To answer this question, we carried out three additional experiments. They are the same as the original sensitivity simulations with prior wind perturbations of negative AO forcing, negative DA forcing and positive BH forcing, respectively, but with the sea surface height $\eta$ in equation (1) replaced with that saved from the control run. In these experiments, the ocean influences sea ice drift only through ice-ocean stress. Technically, the monthly mean sea surface height output from the control run was read in and used in the sea ice momentum equation (1) in these extra simulations. Using monthly mean data saves disk storage space and avoids losing model efficiency compared with doing output/input for every model time step. To see whether using monthly mean sea surface height is sufficient for considering the gradient force for our purpose, we repeated the control simulation and used the monthly mean sea surface height output from the previous control simulation. We found that the difference in the Arctic sea ice relative to the original control simulation is negligible compared with the sea ice anomalies discussed in this paper. We note that the two forcing terms *always co-exist in reality* (because ocean surface geostrophic currents are associated with sea surface height gradient through geostrophic balance) and we only use the three additional simulations for understanding their relative importance.

The sea ice thickness anomalies relative to the control run for the original sensitivity simulations and their counterpart experiments with the sea surface height gradient force taken from the control run are shown in the first and second row of Figure 11, respectively. It is interesting to see that the sea ice thickness anomalies are quite different between the two sets of experiments. Without the perturbations in sea surface height gradient force, strong positive sea ice thickness anomalies are found in a large central area of the Arctic basin in the case of prior negative AO perturbation (Figure 11b), in contrast to the original sensitivity experiment in which positive anomalies are mainly located from north of the Canadian Arctic Archipelago to the eastern Canada Basin (Figure 11a). Outstanding changes are also found in the case with prior positive BH forcing. Instead of negative sea ice thickness anomalies in the Canada Basin (Figure 11g), perturbing the ice-ocean stress alone causes positive anomalies there (Figure 11h). In the case with prior negative DA perturbation, the positive sea ice thickness anomalies have a larger spatial extent without the perturbation in sea surface height gradient force (Figure 11d,e). In all the three cases, eliminating the effect of sea surface height gradient changes the spatial patterns with negative and positive anomalies to patterns dominated by positive anomalies.

The differences in sea ice drift between the two sets of experiments are shown in the bottom row of Figure 11. The drift differences (experiments with perturbations in sea surface height gradient force minus experiments without those perturbations) are orientated roughly from high sea surface height to low sea surface height (compare with Figure 4a,c,e). Their directions and associated sea ice advections are consistent with the differences in sea ice thickness anomalies between the two sets of simulations. In the case with prior negative AO perturbation, those drift anomalies directing from central Arctic towards Canada Basin (Figure 11c) can partially explain the reduced sea ice thickness in the central Arctic and increased thickness north of the Canadian Arctic Archipelago and in the Canada Basin (changing from the result in Figure 11b to that in Figure 11a). In the case of prior positive BH perturbation, the divergent drift anomalies in the Canada Basin (Figure 11i) are consistent with the reduction in sea ice thickness in the Beaufort Gyre (changing from the result in Figure 11h to that in Figure 11g). In the case

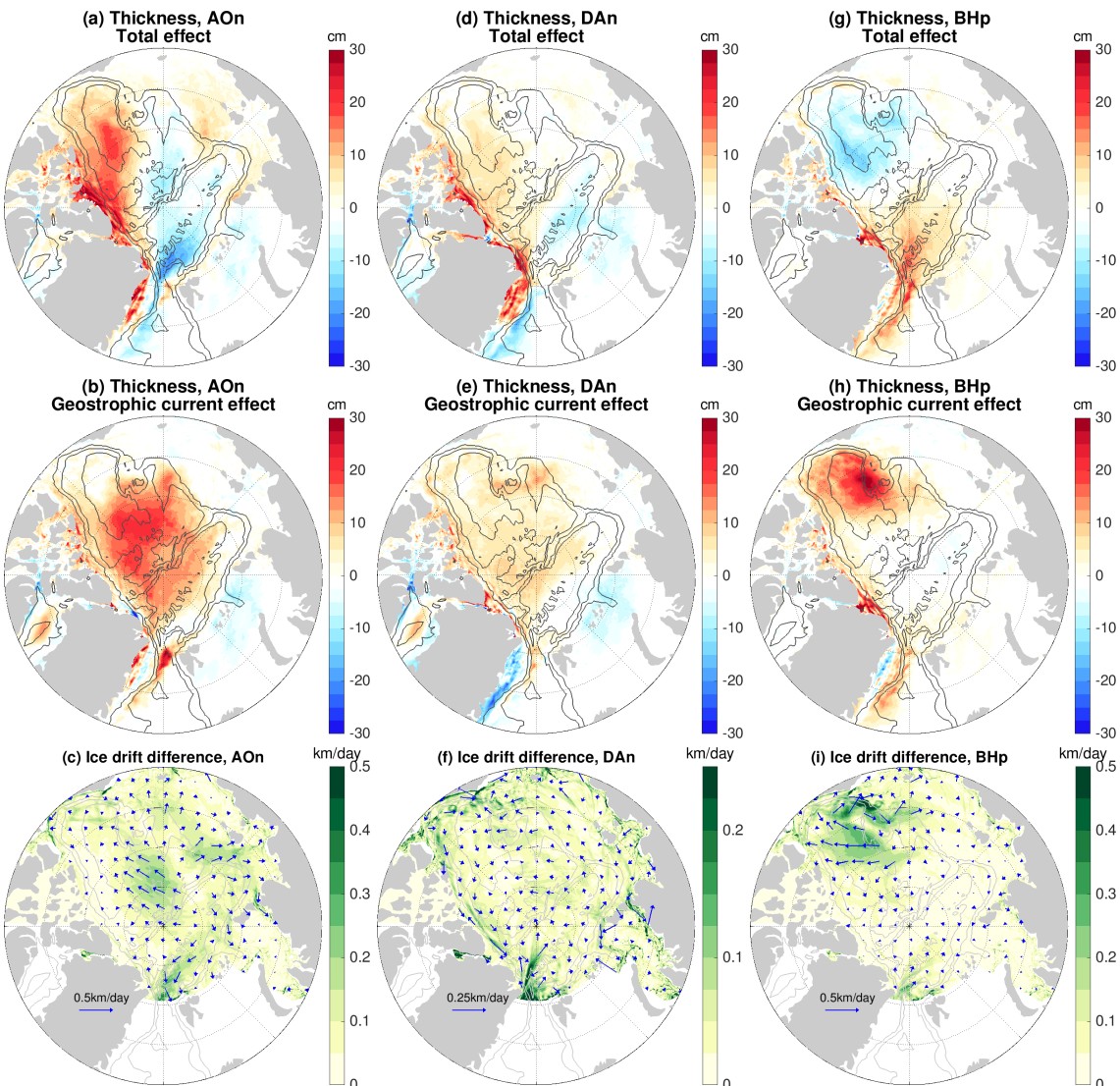

**Figure 11.** (a) Total anomaly of sea ice thickness in the sensitivity experiment with the initial ocean perturbed by applying the negative phase of Arctic Oscillation (AO) perturbation beforehand. The anomaly is referenced to the control run and averaged over the four years of the sensitivity experiment. (b) The same as (a), but for the experiment with the sea surface height gradient force in the sea ice momentum equation taken from the control run, so the result indicates the impact of ocean surface geostrophic current through influencing ocean-ice stress. (c) The difference of sea ice drift between the two experiments associated with (a) and (b). (d)-(f) The same as (a)-(c), but for the experiments with the initial ocean spun up with the negative phase of Dipole Anomaly (DA) perturbation. (g)-(i) The same as (a)-(c), but for the experiments with an initial ocean spun up with the positive phase of Beaufort High (BH) perturbation. Note that the scaling for velocity arrows and color patches in the DA case in (f) is different from other cases in (c) and (i). The black contour lines indicate the 500-, 2000- and 3500-m isobaths.

of prior negative DA perturbation, the drift anomalies directing from eastern Eurasian Basin to western Amerasian Basin and from western Amerasian Basin toward Canadian Arctic Archipelago (Figure 11f) are consistent with the spatial contraction of positive sea ice thickness anomalies (changing from the result in Figure 11e to that in Figure 11d). It is known that the sea ice momentum is mainly balanced between wind stress, ice-ocean stress and sea ice internal force (Steele et al., 1997). Our results supportively show that the sea ice drift anomalies are largely aligned with surface geostrophic current anomalies (Figure 6) and the magnitudes of the drift anomalies associated with sea surface height gradient force are relatively small (up to about 10-20% regionally, cf. Figure 11c,f,i and Figure 6a,c,e). However, these small changes in sea ice drift can produce relatively large differences in sea ice thickness.

We found that the sea ice volume exports through Fram Strait in all sensitivity simulations are not significantly changed relative to the control run (see further discussion below in Section 4.4). Therefore, the predominant positive sea ice thickness anomalies in the simulations without perturbations in the sea surface height gradient force (second row of Figure 11) imply that changes in sea ice thermodynamics have occurred. Indeed, the changes in sea ice thickness due to thermodynamics are predominantly positive and much stronger in these simulations than in the original sensitivity simulations (cf. Figure S9 and Figure 10d,e,f). That is, the impacts of ice-ocean stress alone on sea ice drift would cause overall increases in sea ice thickness in the three considered cases through both dynamic and thermodynamic changes, while the drift anomalies induced by both forcing terms together cause redistribution of sea ice and do not incur significant thermodynamic changes in sea ice. We found that influence of the ocean on sea ice concentration and edge locations also becomes different in the absence of the perturbation in sea surface height gradient force.

In summary, both forcing terms associated with the ocean in the sea ice momentum equation are important in determining the exact sea ice drift anomalies and the associated impact on sea ice thickness and concentration. Both the mean anomalies and the seasonal variation in the magnitudes of the anomalies in sea ice thickness are mainly caused by the changes in sea ice dynamics associated with the changes in sea ice drift induced by the two forcing terms together. The induced thermodynamic changes in sea ice thickness are relatively small.

## 4 Discussion

### 4.1 Initial ocean conditions vs. initial sea ice conditions

In our sensitivity simulations the employed initial sea ice conditions are the same as in the control run, which allows us to easily identify the impact of the ocean memory of prior wind perturbations on sea ice. Actually, the wind perturbations during the model spinup created quite different sea ice states. If we employ sea ice states produced from the wind-perturbation simulations as initial sea ice conditions in the sensitivity simulations, can we still observe the impact of the perturbed ocean states?

We performed one additional experiment, which is the same as the sensitivity simulation with prior negative AO perturbation, but the initial sea ice state is taken from the corresponding wind-perturbation simulation rather than from the control run. The difference of initial sea ice thickness employed in these two experiments is shown on top of Figure 12.

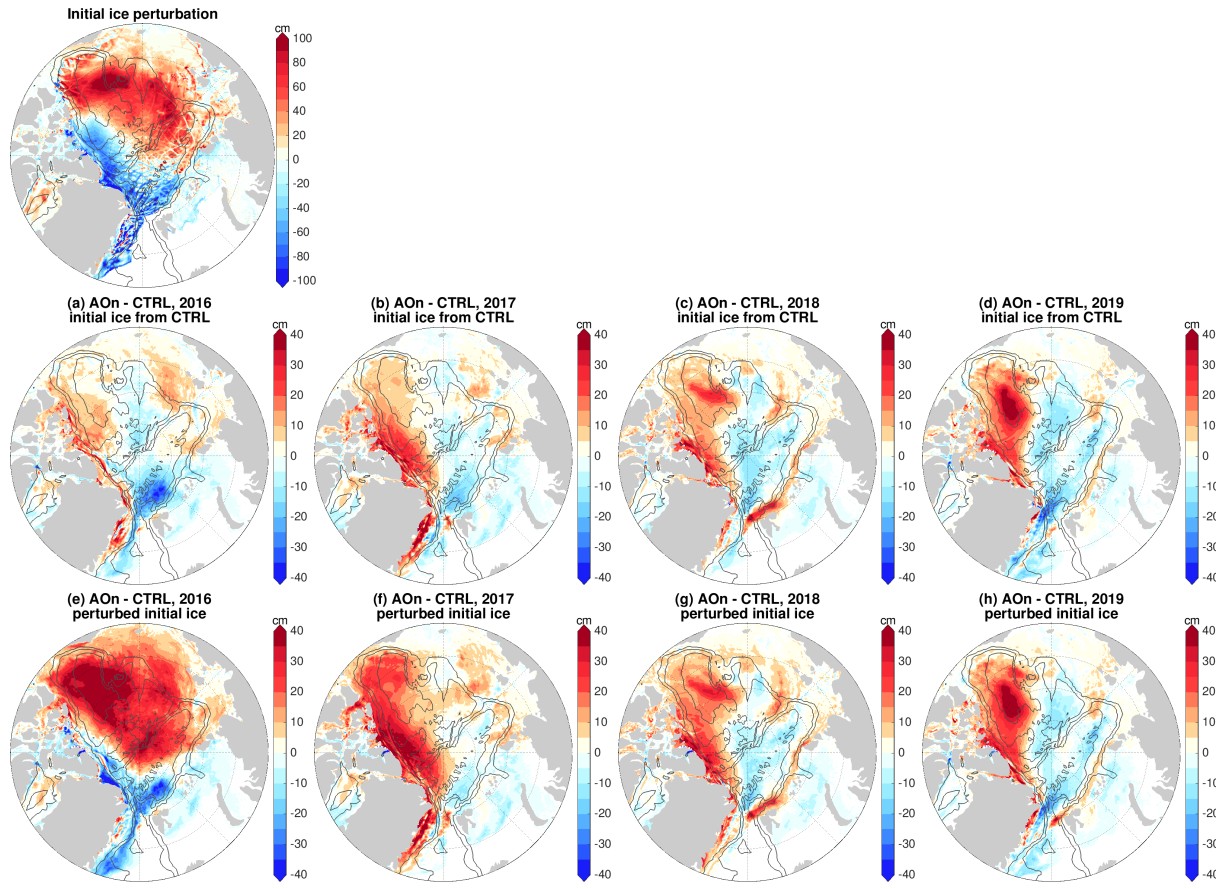

**Figure 12.** Impacts of sea ice initial condition versus ocean initial condition. Anomaly of sea ice thickness in the sensitivity experiment with only the initial ocean perturbed by applying the negative phase of Arctic Oscillation (AO) perturbation beforehand: in (a) 2016, (b) 2017, (c) 2018 and (d) 2019. The anomalies are referenced to the control run. (e)-(h) The same as (a)-(d), but for the experiment with initial sea ice also perturbed. In this experiment, the initial sea ice at the beginning of 2016 is taken from that at the end of the spinup run with negative AO. The difference in the initial sea ice thickness between the two simulations is shown on top of the figure. Comparing (a)-(d) with (e)-(h) indicates that the impact of the initial ocean condition overwhelms the impact of the initial sea ice condition starting from the third year. The black contour lines indicate the 500-, 2000- and 3500-m isobaths.

The annual mean sea ice thickness anomalies relative to the control run for each year in these two simulations are shown in Figure 12a-d and Figure 12e-h, respectively. We see that the anomalies of the two simulations become more similar with time, indicating that the dynamic impact of the ocean becomes more important than the initial perturbations in sea ice. Even with this very large initial difference in sea ice (up to about 1 m in thickness), the sea ice thickness anomalies relative to the control run are already dominated by the impact of the perturbed ocean after two years, and fully represent the ocean impact in the fourth year. This simulation further proves the robustness of our findings on the impact of the ocean's memory. We note that there could be effects from initial sea ice states through potential feedbacks in coupled climate models, which is beyond the scope of this study. Here we used forced sea ice-ocean simulations to reveal the direct impact of the ocean on sea ice.

## 4.2    Impact on sea ice deformation

Changes in sea ice drift, thickness and concentration are associated with changes in sea ice deformation. The deformation rate of pack ice is a key parameter determining the formation of sea ice linear kinematic features like leads (e.g.,  Kwok, 2001; Wang et al., 2016b; Bouchat and Tremblay, 2017; Mohammadi-Aragh et al., 2018, 2020; Hutter et al., 2019; Hutter and Losch, 2020; Rampal et al., 2019). The model resolution we used cannot realistically represent these features due to their narrowness in reality. However, it is interesting to know how the sea ice deformation rates are impacted by the ocean. The anomalies of sea ice deformation rates relative to the control run averaged in non-free-drift seasons (December to June) are shown for the six sensitivity simulations in Figure 13. We found that the ocean perturbations do induce certain changes in the deformation rates depending on cases and regions.

In the control run, larger deformations rates are located in the Beaufort Gyre region and in the marginal seas (top row of Figure 13), consistent with observations (e.g.,  Lindsay et al., 2003). In the cases with prior BH perturbations, strong influence on the deformation rates in the Beaufort Gyre region is found, with up to about 30% changes (Figure 13e,f). The deformation rate anomalies are predominantly positive (negative) in the case of prior positive (negative) BH perturbations. This could be explained by the fact that anticyclonic (cyclonic) ocean circulation strengthens (weakens) the anticyclonic sea ice drift in the Canada Basin. In the cases of prior AO perturbations, the anomalies of the deformation rates are pronounced not only in some marginal sea areas where the deformation rates are large in the control run, but also at some places inside the basin where the deformation rates are relatively low in the control run (Figure 13a,b). Some of these anomalies in the basin have magnitudes similar to the values in the control run, implying their relative importance regionally. The anomaly patterns in the cases of prior negative and positive perturbations are not anticorrelated in all places. This might be associated with the nonlinearity of sea ice deformation and the chaotic nature of linear kinematic features. In the cases of prior DA perturbations, the deformation rate anomalies are smaller than in other cases (Figure 13c,d) as expected from weaker ocean perturbations obtained from DA forcing.

Our results reveal that the impact of the ocean anomalies on sea ice deformation can be relatively strong and suggest that understanding changes in sea ice deformation needs to consider both the atmosphere and ocean forcings. Dedicated studies on the impacts of the ocean on sea ice deformation rates are still required in the future.

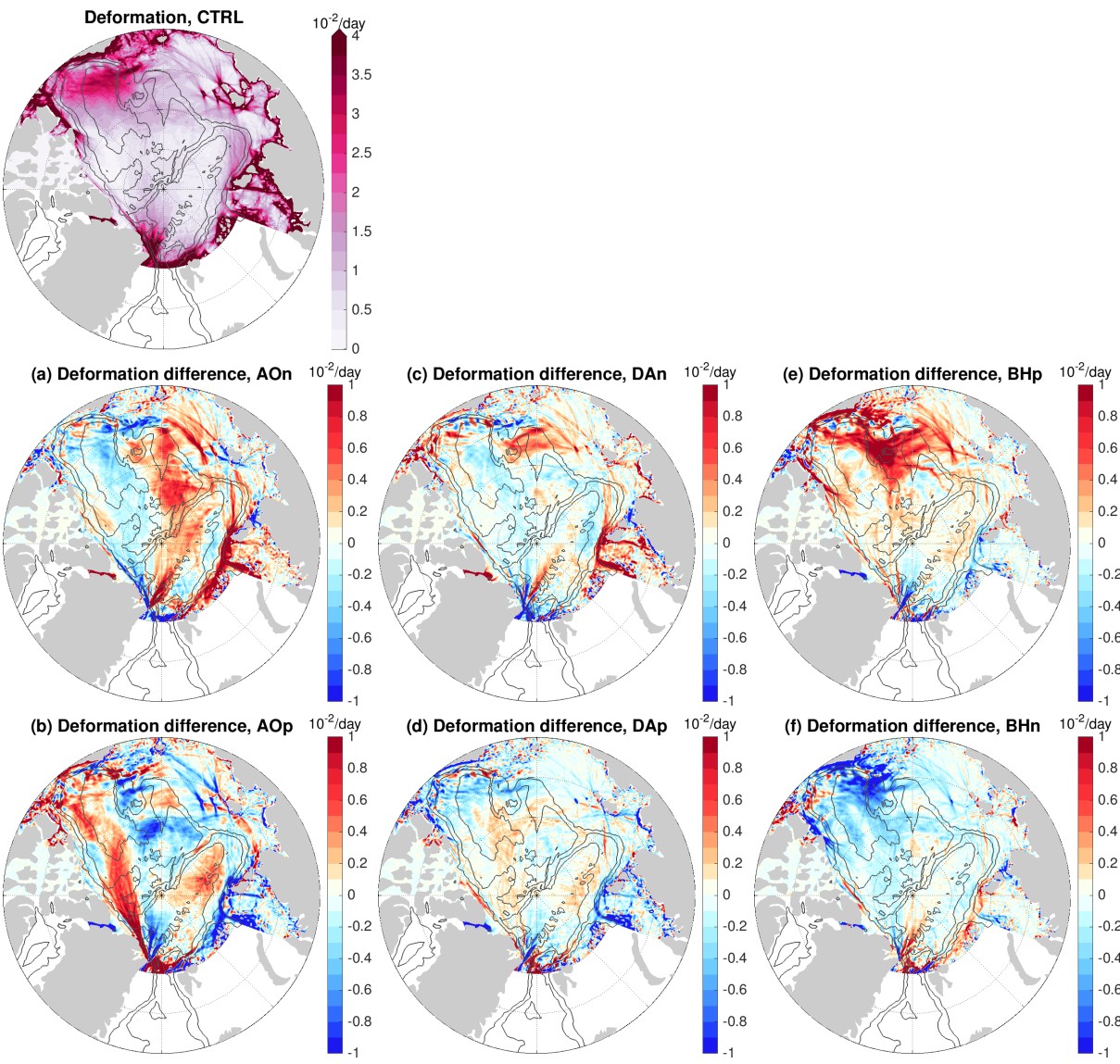

**Figure 13.** (a)-(f) Anomaly of sea ice deformation rate averaged from December to June over the four model years of the sensitivity simulations in which wind perturbations were switched off: Experiments with an initial ocean spun up with (a) negative phase of Arctic Oscillation (AO) forcing, (b) positive phase of AO forcing, (c) negative phase of Dipole Anomaly (DA) forcing, (d) positive phase of DA forcing, (e) positive phase of Beaufort High (BH) forcing, and (f) negative phase of BH forcing. The anomalies are referenced to the control run result, which is shown on top of the figure. The black contour lines indicate the 500-, 2000- and 3500-m isobaths.

## 4.3 Realism of the sensitivity experiments and implications

In this study we used idealized wind perturbations representative of major Arctic atmospheric modes to spin up the ocean for six years before the four-year-long sensitivity simulations. The length of the simulations is within the observed range of typical variation periods of Arctic freshwater content. The amount of liquid freshwater in the Arctic Ocean varied at 5 to 7 year intervals before 1997 and has remained in a state of accumulation for about two decades since then (Proshutinsky et al., 2015). As the changes in liquid freshwater content and sea surface height with positive and negative wind perturbations are roughly antisymmetric (Wang, 2021), the time scale of the recovery of the ocean state after the wind perturbations are switched off, that is, the length of the ocean memory, is expected to be similar to the duration of the prior wind perturbations.

Over the last two decades the Arctic Ocean has accumulated about $11,000 \, \mathrm{km}^3$ extra liquid freshwater, mainly in the Amerasian Basin (Polyakov et al., 2013; Rabe et al., 2014; Wang et al., 2019a; Proshutinsky et al., 2019). In our sensitivity simulations, the perturbed initial ocean has a difference of about $9,000 \, \mathrm{km}^3$, $1,000 \, \mathrm{km}^3$ and $4,000 \, \mathrm{km}^3$ freshwater in the Arctic Ocean relative to the control run (calculated using a reference salinity of 34.8). Therefore, the strength of the sea ice responses to the perturbed ocean obtained in our simulations is plausibly within realistic ranges. In particular, the Beaufort Gyre region had an increase of more than $6,400 \, \mathrm{km}^3$ of liquid freshwater from 2003 to 2018 (Proshutinsky et al., 2019). This implies that the impact of the current ocean states on Arctic sea ice is even stronger than in our sensitivity simulation with prior positive BH perturbation.

We found that the impact of the perturbed ocean on sea ice has strong seasonal variations, which is due to the seasonal variation in sea ice internal stresses. The implication is that the impact of the ocean on sea ice will become stronger in a warming climate as sea ice will become thinner and less compact. It was also found that winds can perturb ocean freshwater content, sea surface height and geostrophic currents more strongly with sea ice decline as compared between the recent sea ice condition and that in the 1980s (Wang, 2021). These effects together will lead to much stronger interannual to decadal variability in sea ice associated with the ocean's memory of wind variability. When we interpret observed sea ice changes, it will become increasingly important to take ocean changes into account.

In our simulations the ocean memory of prior wind perturbations influences sea ice mainly dynamically through the anomalies of sea surface height and surface geostrophic currents. The upper ocean temperature changes caused by winds are small and have much smaller contributions to the obtained sea ice anomalies than the dynamic impact. This is shown by the small sea ice thickness changes associated with sea ice thermodynamics (Figure 10). Furthermore, the dynamic impact can reemerge even though it is masked by applying large sea ice initial perturbations at the beginning (Figure 12). The implication is, even if there is additional initial thermal perturbations that can induce initial sea ice anomalies, the ocean dynamic effect can reemerge as long as the perturbation signals in sea surface height and surface geostrophic currents exist.

We analyzed the dynamic impact of the ocean on sea ice on monthly and longer time scales in this study. As the strength of the impact is very sensitive to the temporal changes in sea ice as revealed by the strong seasonal variability of the impact, regional variability of sea ice internal stress on shorter time scales could also allow for high frequency variability in the impact

of the ocean on sea ice drift. However, on short time scales wind variability is strong and it determines most of the sea ice motion (Thorndike and Colony, 1982), so the impact of the ocean could thus be masked.

The regulation of sea ice states on the impact of the ocean implies that an overestimated sea ice thickness in our model might lead to an underestimation in the induced sea ice changes in the sensitivity experiments. We also note that the idealized wind perturbations we used were intended to allow for easy interpretations of involved dynamical processes. Realistic wind forcing can produce more complicated spatial structures in the response of sea ice.

### 4.4 Indirect vs. direct impacts from winds

The indirect impact of winds on sea ice through the ocean is significant as we revealed in this paper, but it is weaker than the direct impact of wind perturbations when they are present. For comparison, the differences of sea ice thickness, drift, concentration and deformation rates in simulations with the presence of wind perturbations relative to the control run are shown in Figure 14 and Figures S10-S13. The spatial patterns of the direct impact of winds on sea ice could be different from or similar to those induced through the ocean's memory depending on cases and properties investigated. For example, the direct and indirect impacts of BH forcing are similar for the deformation rates in terms of the spatial patterns but quite different for the sea ice thickness. Another example is for the DA forcing, which has a very strong direct impact on sea ice thickness, concentration and area, but only a weak impact through the ocean's memory because of its weak influence on halosteric height.

As mentioned above, the impact of the perturbed ocean on sea ice volume export through Fram Strait is small. The export anomalies are around $100\,\mathrm{km}^3/\mathrm{yr}$ in most cases except that it is about $200\,\mathrm{km}^3/\mathrm{yr}$ in the case of prior negative DA perturbation (Figure S14). In the presence of wind perturbations, the induced changes in sea ice volume export are much larger than those associated with the ocean memory in most forcing cases (Figure S14). The strongest direct impact is in the cases of the DA forcing, reaching about $1,000\,\mathrm{km}^3/\mathrm{yr}$. We note that the locations of the centers of atmospheric circulation modes, which change in time, play a crucial role in determining the sea ice export too (Jung and Hilmer, 2001; Wang et al., 2021a). Therefore, quantitatively, the wind-driven anomalies of the sea ice volume export shown in Figure S14 are only representative of the wind perturbations shown in Figure 2.

## 5 Conclusions

In this numerical study we used wind perturbations to change the Arctic Ocean and then used the perturbed ocean as initial conditions in sensitivity simulations to investigate the impact of wind perturbations existing beforehand on Arctic sea ice. The wind perturbations can change the sea surface height and surface geostrophic currents through changing the ocean's liquid freshwater content, an integral indicator of the upper ocean salinity, thus the halosteric height. In the sensitivity simulations the wind perturbations are turned off, so the previously induced changes in halosteric height, sea surface height and surface geostrophic currents weaken with time, but they can last for years. We found that these lasting changes in the ocean can significantly influence the sea ice drift, thickness, concentration and deformation rates, which manifests a lasting impact of winds on sea ice through the ocean's memory.

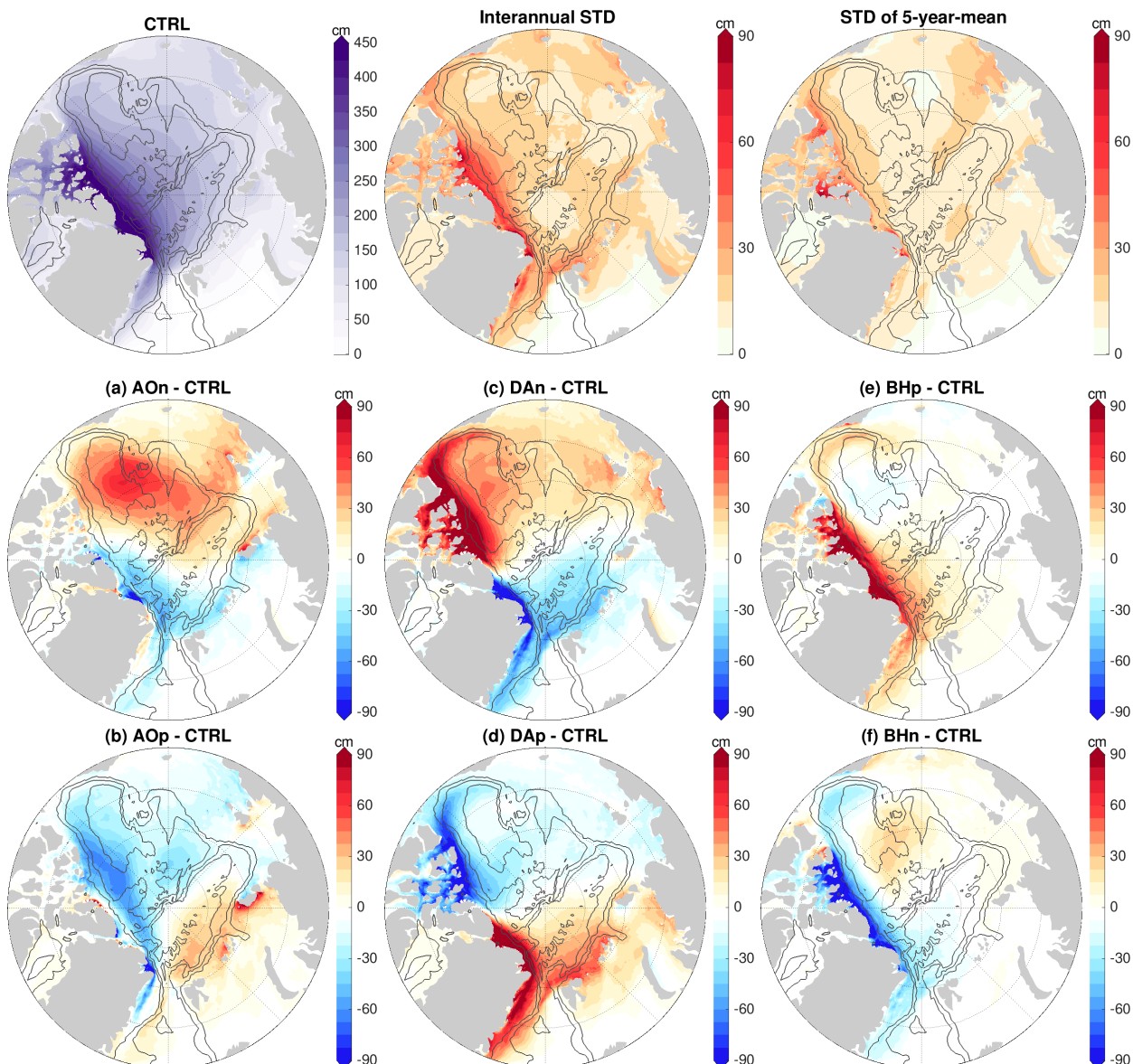

**Figure 14.** (a)-(f) Anomaly of sea ice thickness when wind perturbations are kept in the simulations. Experiments with (a) negative phase of Arctic Oscillation (AO) forcing, (b) positive phase of AO forcing, (c) negative phase of Dipole Anomaly (DA) forcing, (d) positive phase of DA forcing, (e) positive phase of Beaufort High (BH) forcing, and (f) negative phase of BH forcing. The anomalies are averaged over 2016-2019 and referenced to the control run result. The mean sea ice thickness in this period (left), the standard deviation (STD) of sea ice thickness on the interannual time scale in the 2010s (middle), and the STD of the pentadal mean in the period 1980-2019 (right) from the control run are shown on top of the figure for reference. The black contour lines indicate the 500-, 2000- and 3500-m isobaths. This figure shows the impacts of wind perturbations when they are present; It is to be compared with Figure 7 that shows the impacts from the ocean's memory. The direct impacts of winds on sea ice drift, concentration and deformation and their seasonality are shown in the supplementary Figures S10-S13.

The spatial patterns of the changes in sea ice states induced by the perturbed ocean depend on the wind perturbations applied beforehand. We investigated the cases with the ocean perturbed by winds representing Arctic Oscillation (AO), Arctic Dipole Anomaly (DA) and Beaufort High (BH) modes for both their positive and negative phases. In the case of prior negative AO perturbation, sea ice thickness is oppositely changed by the ocean perturbation in two regions. One is the north Canadian Arctic Archipelago and western Canada Basin where sea ice thickness increases, the other is the western Eurasian Basin and central Arctic where it decreases. The changes are opposite in the case of prior positive AO perturbation. In the case of positive (negative) BH perturbation, sea ice thickness is decreased (increased) in the Canada Basin and increased (decreased) in the western Eurasian Basin. In both cases of prior AO and BH perturbations, the ocean, which was perturbed to an extent well within the observed range, can change annual mean sea ice thickness by up to about 10% regionally in our simulations. Compared with the standard deviation of annual and 5-year-mean sea ice thickness, the persistent sea ice thickness anomalies are regionally significant.

In general, the induced sea ice drift anomalies are largely aligned and scaled with the anomalies of ocean surface geostrophic currents. The ocean perturbed with BH forcing can change annual mean sea ice drift speed by up to about 25% in the Beaufort Gyre region, while the ocean perturbed with AO forcing can lead to even stronger relative changes in sea ice drift locally. Compared with the variability of sea ice drift, these drift anomalies are very significant.

The impact of the perturbed ocean on sea ice concentration is pronounced only in summer. The ocean perturbed by AO and BH forcing can moderately influence the location of sea ice edge regionally in the western Arctic in September, with the Arctic September sea ice extent changed by up to about 3-6%. In the cases of prior DA perturbations, the weaker perturbation of the ocean has a weaker impact on sea ice compared to other cases, and the induced changes in sea ice thickness and concentration are not significant compared with the magnitudes of the interannual and pentadal variability. Our simulations further revealed that the ocean can significantly influence sea ice deformation rates.

We found strong seasonal variations in the changes of sea ice drift, thickness and concentration associated with the perturbed ocean. Although the sea ice changes are larger in summer and smaller in winter and spring, they are significant year-round for sea ice drift and thickness. Because the seasonal variations in the ocean (sea surface height and geostrophic currents) perturbations are negligible, the seasonal variations in the sea ice changes can only be explained by the seasonal variability of sea ice internal stress. Weaker internal stress allows the impact of the ocean to be more pronounced. This implies that the impact of the ocean on sea ice in the future warming climate will become stronger.

Our analysis revealed that the impact of the perturbed sea surface height and surface geostrophic currents on sea ice is mainly through changing sea ice drift and sea ice dynamics, not sea ice thermodynamics. We also found that both the changes in ice-ocean stress and sea surface height gradient force, two of the sea ice momentum budget terms, are important in determining the exact changes in sea ice. The changes in sea ice thickness would be fully different without the subtle role of sea surface height gradient force.

The strong impact of ocean dynamic changes on sea ice suggests that not only the changes in the atmosphere should be considered for understanding observed changes in sea ice, but also the changes in the ocean and the changes in the response of the ocean to the atmosphere should be taken into account. Sea ice drift pathways changed significantly in different years even

for ice floes originating from the same location (Krumpen et al., 2021). The spatial distribution of liquid freshwater and ocean surface circulation in the Arctic Ocean have varied considerably over the recent decades (e.g., Timmermans et al., 2011; Wang et al., 2019c; Polyakov et al., 2020), which could have contributed to the changes in sea ice drift pathways, besides the direct impact of winds. Our finding also addresses that the changes in ocean surface geostrophic currents must be taken into account in the calculation of ice-ocean stress. For example, the impact of ocean circulations on ice-ocean stress can strongly influence the estimate of sea ice drag coefficients (Heorton et al., 2019) and the ocean surface Ekman pumping which determines the spinup/spindown of the Beaufort Gyre (Dewey et al., 2018; Zhong et al., 2018; Meneghello et al., 2018; Wang et al., 2019a). The on-going sea ice decline can significantly strengthen the ocean response to winds (Davis et al., 2014; Martin et al., 2016; Wang et al., 2019c, 2021b). The ocean circulation with stronger variability in a warming climate (Wang, 2021) could thus more strongly affect the sea ice variability, which should be considered when explaining observed sea ice changes. Our study further suggests the importance of the initialization of ocean dynamics, especially in terms of ocean salinity, for Arctic regional sea ice predictions on seasonal to decadal time scales. Both ocean and sea ice initializations were found to be important for improving the prediction of Arctic sea ice and climate (e.g., Tian et al., 2021). Previous analysis of the role of ocean initialization has been mainly focused on the impact of ocean temperature. The role of salinity initialization for polar prediction through its dynamic impact still needs to be further investigated.

*Code and data availability.* The FESOM model version used in this study is available at https://doi.org/10.5281/zenodo.1116851. The sea ice model results are available at https://doi.org/10.5281/zenodo.4509500.

*Author contributions.* QW conceived this study, performed the simulations and wrote the first draft of the paper. All coauthors contributed to the data processing, model result interpretation and paper improvement.

*Competing interests.* The authors declare that they have no conflict of interest.

*Acknowledgements.* The authors thank the reviewers (Céline Heuzé and Leandro Ponsoni) and the editor (Michel Tsamados) for their helpful comments. This work was supported by the German Helmholtz Climate Initiative REKLIM (Regional Climate Change), by the FRontiers in Arctic marine Monitoring program (FRAM), by the German Federal Ministry for Education and Research (BMBF) within both the GROCE2 project (Grant number 03F0855A) and the EPICA project (Grant number 03F0889A), and by the Collaborative Research Centre TRR 181 "Energy Transfer in Atmosphere and Ocean" funded by the Deutsche Forschungsgemeinschaft (DFG, German Research Foundation) with Projektnummer 274762653.

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
