# Peer review of "Lasting impact of winds on Arctic sea ice through the ocean's memory"

_The Cryosphere, 2021_

## Referee Comment (RC2)

Dear Authors, Editor,

This is my first review of the manuscript *"Lasting impact of winds on the Arctic sea ice through the ocean's memory"* by Q. Wang and collaborators. In this study, the authors show interesting results based on the fact that the ocean "stores" the impact of different patterns of atmosphere circulation (Arctic Oscillation, Arctic Dipole Anomaly, and Beaufort High modes) which, a few years after, will influence the Arctic sea ice drift, thickness, concentration, and extent. The authors argue that the ocean impact on sea ice occurs mainly through changes in the sea ice dynamics (compared to the thermodynamics). These, in turn, are forced by changes in the sea surface height and geostrophic ocean currents. I think the numerical experiments described in Sec. 2 are well-thought and -designed. Although, I am not convinced of the meaning of the experiments described in Sec. 3.3 (please see below). Overall, the text is well written and pleasant to read, so that I do not have many "line-by-line" comments.

Since I am late with my review (for which I would like to apologize), it is worthwhile saying that I have seen that the first referee already plotted the comments online. To avoid getting biased by those comments, I haven't look at them yet. Thus, regarding this aspect, my suggestions are independent.

In short, I think the manuscript is nearly ready. I encourage its publication. However, I do have three major comments that the authors might consider addressing and/or answering in case I have missed the point.

**Major comments:**

**1.** Since the authors propose to look at the ocean's memory, I feel that the study could provide further information on the memory's timescale. For how long the memory imposed by the different atmospheric modes is stored by the ocean? And, for how long it will impact the sea ice? In practical terms, how many years more are required for the lines displayed in Fig. 5a,b,c to converge to zero? I understand that this would require extra modeling effort what is not always straightforward (or even feasible), but I think the scientific community would benefit very much from that information. Interesting conclusions could be achieved. Among others, this effort can lead to findings such as "the negative phase of atmospheric mode X remains longer in the ocean compared with its positive phase (or the other way around)". Or, "the AO forcing has a higher and longer-lasting impact on SSH anomaly compared to the BH forcing". Etc.

**2.** The authors argue that sea ice changes take place through sea ice dynamics. However, I guess they can provide a more comprehensive analysis regarding the thermodynamics aspect. For instance, by looking at other diagnostics such as providing a comparison in terms of ocean heat content between control and sensitivity experiments, and inspecting its relation with the sea ice changes.

**3.** I am not convinced that the experiments described in Sec. 3.3 effectively disentangle the "ice-ocean stress" and "sea surface height gradient force" contributions. Since the geostrophic flow, and therefore the ice-ocean stress, is generated by gradients in the sea surface height, aren't the second (ice-ocean

stress) and third (sea surface height gradient force) terms on the right-hand-side of the Eq. 1 intrinsically related?

In other words, I am wondering whether statements such as in **pg. 17, ls. 255–258**

*"They are the same as the original sensitivity simulations with prior wind perturbations of negative AO forcing, negative DA forcing and positive BH forcing, respectively, but with the sea surface height η in equation (1) replaced with that saved from the control run. In these experiments, the ocean influences sea ice drift __only__ through ice-ocean stress."*

make sense since this change is impacting the geostrophic balance (at least the barotropic component). I mean, the geostrophic currents are a consequence of the horizontal gradients of sea surface height. What is the physical meaning of introducing a geostrophic circulation which is not in balance with the corresponding sea surface height? I have the feeling that the model will quickly adjust the geostrophic currents to the new sea surface height. I might be missing something, but I can't see the meaning of these experiments. Could the authors say a few words on that?

**Overall comment on the experiments' description:**
I found a bit confusing that the three main sensitivity experiments are described in Sec. 2 (Method and model setups) while the "additional experiments" are described in Sec. 3.3 (Attribution of the impact). While reading the manuscript, I wasn't expecting new experiments "jumping" into the text when discussing the results. This comment might be biased by my personal taste, but I think that all experiments could be described upfront in Sec. 2. I leave that to the authors.

**pg. 1, l. 9:** "We identified" → it seems that "identified" isn't the right term here. Maybe "reproduced"?

**pg. 2, l. 22:** *"pronounced interannual and multiyear variability"* → "interannual" and "multiyear" sounds kind of the same.

**pg. 4, ls. 72–73:** *"climatological sea ice derived from a previous simulation."* → a few more details on the "previous simulation" would be welcome.

**pg. 4, ls. 75–76:** *"The mean values of Arctic sea ice volume and extent are slightly overestimated"* → By curiosity, do the authors expect impact of this overstimation on their results? If so, what impacts?

**pg. 4, l. 86:** *"deseasonalized monthly mean sea level pressure"* → how did the authors deseasonalized the time series? Filtering, subtracting the seasonal and/or monthly means?

**pg. 4, l. 94:** *"Wind anomalies associated with three idealized SLP anomalies representing these modes were used:"* → Also out of curiosity, how the idealized fields were created? I see that the idealized fields (Fig. 2b,e,h) do not perfectly match the modes shown in (Fig 2a,d,g)? For instance, maximum values are not co-collocated in space. Do the authors expect that these differences (even if small) impact the sea ice concentration patterns shown in Fig. 8 and the sea ice extent, for instance?

**pg. 6, ls. 107–109:** *"The main dynamical processes changing Arctic freshwater content under the wind perturbations are __Ekman transport of freshwater__, although induced changes in sea ice thermodynamics also have certain contributions"* → Even underneath the sea ice where the wind stress isn't applied to the sea surface?

**pg. 10, Fig. 6's caption:** Maybe it is worth to mention that the scaling for blue and red velocity arrows is the same (If that is the case).

**Leandro Ponsoni**
**28/May/2021**

---

## Author Comment (AC1)

**Reply to Reviewer #1**

Dear Dr. Céline Heuzé,

Thank you very much for reading our manuscript and providing very useful suggestions. We did the revision according to your and the other reviewer's comments and gave explanations in case we did not do it. See our detailed replies below (in blue).

I'll be brief: Interesting paper that would fit very well within this journal, but some crucial methodological points are currently missing.
Major comments, by order of appearance in the text:

1.      The temporal resolution of the forcings and of the model itself are never discussed. Monthly and annual results are shown, so I assume "monthly" was the highest frequency analysed. If so, the authors should at least discuss how different the results would be with daily/5-daily output (thinking of sea ice drift in particular).

We added a short discussion on possible impact of the ocean on sea ice on shorter time scales in Section 4.3 (lines 391-394): "We analyzed the dynamic impact of the ocean on sea ice on monthly and longer time scales in this study. As the strength of the impact is very sensitive to the temporal changes in sea ice as revealed by the strong seasonal variability of the impact, regional variability of sea ice internal stress on shorter time scales could then allow for high frequency variability in the impact of the ocean on sea ice drift, thickness and concentration". We also added the information on the temporal resolution of the model and atmosphere forcing in the method section (lines 71, 73-75).

2.      Likewise, I am missing a discussion on whether the run lengths are sufficient. Is a 6-year perturbation enough? Is a 4-year period without forcing enough? Or are we, throughout the paper, looking at transient responses?

The Arctic Ocean in reality varies with atmospheric forcing, not being in a steady state. In Discussion Section 4.3, we provided some background on the observed natural variability of Arctic freshwater content (5 - 7 years) and the recent trend in the last two decades. As the period of the variability changes with time in reality, we just choose a reasonable perturbation length for dynamics understanding, the main purpose of this study.

3.      [my biggest issue] The relevance of liquid freshwater, instead of e.g. integrated salt content, is more and more debated within the physical oceanography community. This is particularly problematic for a Pan-Arctic study, to investigate a change that will impact the salinity, using a model. So first, please quantify your model's (potential) biases in upper-ocean salinity. Furthermore, you're not integrating over a fixed depth but only until the depth of the reference salinity. Again, show how well that depth is represented in your model AND how the depth changes throughout the basin and throughout your runs. Now, taking both comments into account, I'd also like to see an evaluation of the robustness of your results by comparing isohaline

vs fixed depth, and reference salinity vs integrated salt content. It can be as simple as showing maps of the mean FWC in the control run for all four options.

To avoid concern, we replaced the freshwater content with halosteric height in the figures (Figure 3 and Figure S3).

The changes in sea surface height can be explained by the changes in halosteric height, as shown by the comparison between figure 4 and figure 3. The latter (changes in halosteric height) are associated with the changes in freshwater content. The very good spatial correlation of freshwater content (was shown in the old paper version) and sea surface height is a direct illustration. As showing halosteric height already serves the purpose, we do not show freshwater content in figures in the revised version.

4.      You never explain how sea surface height is computed. In particular, is it produced by the model, or did you have to compute it afterwards e.g. from the temperature and salinity fields?

The sea surface height is taken directly from the simulations. We added in line 111 "(the dynamical sea level simulated by the model)". It is different from the term "steric height".

5.      [my second biggest issue] As per point 3, I would like to see an assessment of the model's sea ice concentration, thickness, drift speed and seasonal cycle in the control run before you move to investigating the potential differences coming from the different forcings.

We notice that your concern is due to missing seasonality in the "SSH anomaly" from your specific comments below. As we replied below, the SSH does have strong seasonality in each simulation, although the SSH difference between the sensitivity runs and the control run does not have significant seasonality. Therefore, there is no issue with the simulated seasonality of related fields in the model.

Anyway, we added a few sea ice assessments below (Figures R1-1, R1-2, R1-3). Sea ice concentration, thickness and drift were reasonably reproduced. Sea ice thickness is overestimated north of Greenland, but the model results are within the observation uncertainty range in most of the areas. We added Figure R1-3 to the online supporting information (SI, Figure S1). The original Figure S2 already shows integral assessment of sea ice (as extent and volume), so we did not further add Figures R1-1 and R1-2 to the paper.

[Figure]

Figure R1-1. (upper) Simulated sea ice concentration in (a) March and (b) September. (middle) OSI-SAF observed sea ice concentration in (c) March and (d) September. The average over 2000-2019 is shown. The total observational uncertainty is shown in (e) for March and (f) for September. Observation reference: Lavergne, T., Sørensen, A. M., Kern, S., Tonboe, R., Notz, D., Aaboe, S., Bell, L., Dybkjær, G., Eastwood, S., Gabarro, C., Heygster, G., Killie, M. A., Brandt

Kreiner, M., Lavelle, J., Saldo, R., Sandven, S., and Pedersen, L. T.: *Version 2 of the EUMETSAT OSI SAF and ESA CCI sea-ice concentration climate data records*, The Cryosphere, 13, 49-78, doi:10.5194/tc-13-49-2019, 2019.

[Figure]

Figure R1-2. (a) Simulated and (b) CryoSat-2 observed sea ice thickness averaged over months with observations available (October to April) from 2011 to 2019. (c) The CryoSat-2 observation uncertainty. Observation reference: Hendricks, S. and Ricker, R. (2019): Product User Guide & Algorithm Specification: AWI CryoSat-2 Sea Ice Thickness (version 2.2), hdl:10013/epic.8eb07093-4042-40ab-bfb8-e0c72c1567de

**Winter mean (Oct-Apr) 2010-2018**

[Figure]

Figure R1-3. Sea ice drift in OSI-SAF observation and model simulation. The average is taken over cold seasons (October to April) in the period of 2010-2018. Observation is more accurate in the cold season (shown here) than in summer time (Lavergne et al., 2010). Reference: Lavergne, T., Eastwood, S., Teffah, Z., Schyberg, H. and L.-A. Breivik (2010), Sea ice motion from low resolution satellite sensors: an alternative method and its validation in the Arctic. *J. Geophys. Res.*, 115, C10032, doi:10.1029/2009JC005958.

6.      Overall, there are many instances of inconsistencies or lack of precision that force the reader to guess what you really are showing. I am giving examples in the next part of this review, but throughout the manuscript verify that your text, captions and colorbars are not showing different things (velocities vs direction, anomalies vs actual values).

We did the proofreading and addressed your specific comments as replied below.

Specific comments:

●      line 74: liquid freshwater content is only defined line 98. At least say here that the definition comes below.

As replied above, we do not show freshwater content in figures any more and the definition is thus removed. Only in the discussion section where we discuss recent changes in freshwater content, we mentioned the reference salinity in calculating freshwater content from the model results.

● line 86 – 97: this paragraph falls out of nowhere. Start line 86 with e.g. "to design the perturbations, we look at..."

We added "The wind perturbations were designed as described below." (L90)

● Figs 3 and 4: show the control as well – as you did for Fig 6.

As replied above, we replaced freshwater content with halosteric height in figure 3 now, which has no sense with showing its mean value. Furthermore, the standard deviation on interannual and 5-yr time scales in Figure 6 is just provided to show how significant the induced sea ice changes are compared with the strength of the variability on these time scales, while there is no such motivation in figure 3 and 4. The variability of sea surface height was found to be informative only after decomposition using for example EOF analysis (Koldunov et al., 2015; Xiao et al., 2020), which is not the focus in the current paper.

Ref:

Koldunov, N. V., Serra, N., Köhl, A., Stammer, D., Henry, O., Cazenave, A., Prandi, P., Knudsen, P., Andersen, O. B., Gao, Y., and Johannessen, J.: Multimodel simulations of Arctic Ocean sea surface height variability in the period 1970–2009, Journal of Geophysical Research: Oceans, 119, 8936–8954, 2014.

Xiao, K., Chen, M., Wang, Q., Wang, X., and Zhang, W.: Low-frequency sea level variability and impact of recent sea ice decline on the sea level trend in the Arctic Ocean from a high-resolution simulation, Ocean Dynamics, 70, 787–802, 2020.

● Fig 5: I suspect that the lack of seasonality of SSH is either the result of inadapted axis limits (giving the values in the text would help) or potentially too high a sea ice cover year-round (see major point 5).

Fig 5 shows the SSH difference between the perturbed runs and the control run, not SSH in an individual run. SSH in each individual run does have clear seasonality (see Figure R1-4 below). The SSH difference between the runs reflects the lasting ocean memory of prior wind perturbations. We added this information in the figure caption of Fig. 5 in the revised version to avoid confusion.

[Figure]

Figure R1-4: Monthly mean SSH in the control run averaged in different regions. The anomaly referenced to the respective mean value is shown.

●      Fig 6: what is meant by "anomaly in drift or current", as in, what is subtracted from what? It would be clearer to show the arrows of each experiment, and let the reader compare to the control panel. And shading that has only positive values suggests that all experiments have a faster drift than control (shading described as an anomaly, see also major point 6)

Arrows show the anomalies of vectors (ice drift and ocean current), referenced to the control run as mentioned in the figure caption. The anomalies of ice drift and ocean current are shown together to illustrate the impact of the ocean current anomaly on sea ice drift anomaly. This information would be less obvious if we show the vectors of each simulation, and the plots would also become too busy with four arrows put together.

●      It is confusing that Fig 7 shows thickness but Fig 8 shows concentration, when Fig 8 is presented as the seasonal version of the Fig 7 discussion. Potentially show thickness only, or combine both diagnostics into sea ice volume.

Both the impacts on sea ice thickness and concentration are interesting, so we would like to show them separately. However, the impact on sea ice concentration is mainly along the ice edge and in summer. If we show the temporal mean in Figure 8, this information will be masked by small impacts in other seasons and by (seasonal and interannual) changes in sea ice edge locations. We added in the revised paper: "Because the impact on sea ice concentration is only significant in summer and close to sea ice edges, whose locations vary strongly in time, averaging the sea ice concentration anomalies over the four model years would mask the ocean impact. Therefore, in Figure 8 we showed the September sea ice concentration anomaly in one particular year." (lines 195-198)

●      Fig 9: RMS of drift = specify that it is of the drift speed

Added in the figure caption.

● line 257: show this SSH saved from the control run, including its seasonal cycle.

As shown in Figure R1-4 above, there is clear seasonality in the SSH in an individual run.

● line 295: show this result (that the sea ice volume export through Fram Strait is not impacted by sensitivity experiments), that's an important one.

In Discussion Section 4.4 we mentioned the results in more detail and the figure is actually shown as an SI figure (Figure S12). At the place of the original line295 (in line 305 in the revised version), we add "(see further discussion below in Section 4.4)".

Sincerely,
The authors

---

## Author Comment (AC2)

**Reply to Reviewer #2**

Dear Dr. Leandro Ponsoni,

Thank you very much for reading our manuscript and providing very useful suggestions. We did the revision according to your and the other reviewer's comments and gave explanations in case we did not do it. See our detailed replies below (in blue).

This is my first review of the manuscript "Lasting impact of winds on the Arctic sea ice through the ocean's memory" by Q. Wang and collaborators. In this study, the authors show interesting results based on the fact that the ocean "stores" the impact of different patterns of atmosphere circulation (Arctic Oscillation, Arctic Dipole Anomaly, and Beaufort High modes) which, a few years after, will influence the Arctic sea ice drift, thickness, concentration, and extent. The authors argue that the ocean impact on sea ice occurs mainly through changes in the sea ice dynamics (compared to the thermodynamics). These, in turn, are forced by changes in the sea surface height and geostrophic ocean currents. I think the numerical experiments described in Sec. 2 are well-thought and -designed.

Although, I am not convinced of the meaning of the experiments described in Sec. 3.3 (please see below). Overall, the text is well written and pleasant to read, so that I do not have many "line-by-line" comments.

Since I am late with my review (for which I would like to apologize), it is worthwhile saying that I have seen that the first referee already plotted the comments online. To avoid getting biased by those comments, I haven't looked at them yet. Thus, regarding this aspect, my suggestions are independent.

In short, I think the manuscript is nearly ready. I encourage its publication. However, I do have three major comments that the authors might consider addressing and/or answering in case I have missed the point.

Thank you for your comments. See our replies below.

Major comments:
1. Since the authors propose to look at the ocean's memory, I feel that the study could provide further information on the memory's timescale. For how long the memory imposed by the different atmospheric modes is stored by the ocean? And, for how long it will impact the sea ice? In practical terms, how many years more are required for the lines displayed in Fig. 5a,b,c to converge to zero? I understand that this would require extra modeling effort what is not always straightforward (or even feasible), but I think the scientific community would benefit very much from that information. Interesting conclusions could be achieved. Among others, this effort can lead to findings such as "the negative phase of atmospheric mode X remains longer in the ocean compared with its positive phase (or the other way around)". Or, "the AO forcing has a higher and longer-lasting impact on SSH anomaly compared to the BH forcing". Etc.

As the changes in freshwater content and sea surface height with positive and negative wind perturbations are *roughly* antisymmetric (see Figure R2-1 below), the time scale of the recovery of the ocean state after the wind perturbations are switched off, that is, the length of the ocean memory, is expected to be similar to the duration of the prior wind perturbations. We added this short discussion on time scales to the revised paper (lines 366-368). In terms of the magnitude of the impact on sea ice, AO and BH perturbations are more important because they can induce larger changes in freshwater content and sea surface height, which is clear in the paper.

[Figure]

Figure R2-1: Changes in freshwater content in response to different wind perturbations and in different climate scenarios. The implication of these plots is that the time scale of the recovery of the ocean state after the wind perturbations are switched off is close to the duration of the prior wind perturbations. This figure is taken from Wang (2021) cited in the paper.

2. The authors argue that sea ice changes take place through sea ice dynamics. However, I guess they can provide a more comprehensive analysis regarding the thermodynamics aspect. For instance, by looking at other diagnostics such as providing a comparison in terms of ocean heat content between control and sensitivity experiments, and inspecting its relation with the sea ice changes.

Changes in temperature does not tell whether sea ice is indeed impacted by the ocean thermodynamically, while sea ice "thermodynamic growth rate" (the changing tendency of sea ice thickness) shown in the paper explicitly tells the effect of the ocean memory. In particular, the

thermodynamic growth rate in Figure 10 shows to be relatively small anyway, so we do not need more attribution.

3. I am not convinced that the experiments described in Sec. 3.3 effectively disentangle the "ice-ocean stress" and "sea surface height gradient force" contributions. Since the geostrophic flow, and therefore the ice-ocean stress, is generated by gradients in the sea surface height, aren't the second (ice-ocean stress) and third (sea surface height gradient force) terms on the right-hand-side of the Eq. 1 intrinsically related?
In other words, I am wondering whether statements such as in pg. 17, ls. 255–258
"They are the same as the original sensitivity simulations with prior wind perturbations of negative AO forcing, negative DA forcing and positive BH forcing, respectively, but with the sea surface height η in equation (1) replaced with that saved from the control run. In these experiments, the ocean influences sea ice drift only through ice-ocean stress."
make sense since this change is impacting the geostrophic balance (at least the barotropic component). I mean, the geostrophic currents are a consequence of the horizontal gradients of sea surface height. What is the physical meaning of introducing a geostrophic circulation which is not in balance with the corresponding sea surface height? I have the feeling that the model will quickly adjust the geostrophic currents to the new sea surface height. I might be missing something, but I can't see the meaning of these experiments. Could the authors say a few words on that?

We only modified the "sea ice" momentum equation in the sea ice model, not in the ocean model. So the geostrophic currents and SSH remained consistent in the ocean model.

These extra experiments are only intended to disentangle the two forcing terms in the sea ice momentum equation. Yes, these two forcing terms co-exist in reality, as we also addressed in the paper text. But we are interested to know whether both forcing terms are important.

Overall comment on the experiments' description:
I found a bit confusing that the three main sensitivity experiments are described in Sec. 2 (Method and model setups) while the "additional experiments" are described in Sec. 3.3 (Attribution of the impact). While reading the manuscript, I wasn't expecting new experiments "jumping" into the text when discussing the results. This comment might be biased by my personal taste, but I think that all experiments could be described upfront in Sec. 2. I leave that to the authors.

Now we briefly introduced the extra experiments in the Method section (L102-107), while keeping more details where these experiments are analyzed. In the Method section the motivation to do these experiments is less clear, so we provide more information just before showing their results.

pg. 1, l. 9: "We identified" → it seems that "identified" isn't the right term here. Maybe "reproduced"?
Changed to "obtained"

pg. 2, l. 22: "pronounced interannual and multiyear variability" → "interannual" and "multiyear" sounds kind of the same.
Changed to "variability on different time scales" (L22)

pg. 4, ls. 72–73: "climatological sea ice derived from a previous simulation." → a few more details on the "previous simulation" would be welcome.
We added the explanation: "that is, December sea ice averaged over 1970 - 1990 obtained from a simulation with the same model configuration" (L76)

pg. 4, ls. 75–76: "The mean values of Arctic sea ice volume and extent are slightly overestimated" → By curiosity, do the authors expect impact of this overstimation on their results? If so, what impacts?
We added a short comment that model biases could influence the quantitative results. "The regulation of the sea ice state on the impact of the ocean implies that an overestimated sea ice thickness in our model might lead to an underestimation in the induced sea ice changes in the sensitivity experiments. We also note that the idealized wind perturbations we used were intended to allow for easy interpretations of involved dynamical processes. Realistic wind forcing can produce more complicated spatial structures in the response of sea ice." (L395-398)

pg. 4, l. 86: "deseasonalized monthly mean sea level pressure" → how did the authors deseasonalized the time series? Filtering, subtracting the seasonal and/or monthly means?
The mean seasonal cycle was removed. (L91)

pg. 4, l. 94: "Wind anomalies associated with three idealized SLP anomalies representing these modes were used:" → Also out of curiosity, how the idealized fields were created? I see that the idealized fields (Fig. 2b,e,h) do not perfectly match the modes shown in (Fig 2a,d,g)? For instance, maximum values are not co-collocated in space. Do the authors expect that these differences (even if small) impact the sea ice concentration patterns shown in Fig. 8 and the sea ice extent, for instance?
The idealized winds are intended to simplify the study, providing key information without being bothered by small details. As an example one can find more details about designing idealized wind forcing in Marshall et al. (2017) that is cited in our paper. In the revised paper, we added a short comment together with the possible impact of model biases mentioned in the reply above. (L395-398)

pg. 6, ls. 107–109: "The main dynamical processes changing Arctic freshwater content under the wind perturbations are Ekman transport of freshwater, although induced changes in sea ice

thermodynamics also have certain contributions" → Even underneath the sea ice where the wind stress isn't applied to the sea surface?

Yes, wind influences the sea ice directly, also the ocean through changing ocean-ice stress.

pg. 10, Fig. 6's caption: Maybe it is worth mentioning that the scaling for blue and red velocity arrows is the same (If that is the case).

In the figure caption we mentioned "in each panel the same scaling is used for sea ice drift and geostrophic current"

Sincerely,
The authors

---

## Author Response (AR1)

**Dear Editor,**

Thank you for your helpful comments. We revised the paper following the comments from you and the two reviewers. See our detailed replies below and in the reply letters to the reviewers.

- Your introduction is partial and I encourage you to perform more literature research. At present several important papers are missing from authors such as G Meneghello, C. Lique, H. Johnson, P. Davies, T. Martin, H. Heorton etc...I think that a reference to the seminal work by Thorndike is also relevant here.

We added a few important citations closely related to the main topic of the paper in the introduction.

"The ocean surface current and sea ice drift are strongly coupled through the ice-ocean stress (e.g., Tsamados et al., 2014; Heorton et al., 2019). Sea ice has strong internal stress seasonally, so its state can significantly influence the ocean circulation through changing the ice-ocean stress (Martin et al., 2016). In the Beaufort Gyre region, sea ice can limit the spinup of the ocean circulation when ocean surface geostrophic velocity exceeds sea ice drift (Dewey et al., 2018; Zhong et al., 2018; Meneghello et al., 2018; Wang et al., 2019a). As Arctic sea ice declines, the response of sea ice drift to wind variability intensifies, which can thus strengthen the variability of the Arctic sea surface height and surface geostrophic currents (Wang, 2021). A significant part of the sea ice motion averaged over several months can be due to ocean surface currents (Thorndike and Colony, 1982). It was found that the spinup of the ocean can accelerate sea ice drift in the Beaufort Gyre region, especially in warm seasons when sea ice internal stress is low (McPhee, 2013; Kwok and Morison, 2017; Wang et al., 2019a), which may cause sea ice export from the gyre (McPhee, 2013)." (Lines 37-46)

"The Arctic liquid freshwater content varies on a quasi-decadal time scale as a memory of wind forcing (Proshutinsky et al., 2015; Johnson et al., 2018)." (Line 29)

"Due to the combination of a dominant anticyclonic wind regime, enhanced momentum transfer resulting from sea ice decline anda freshening of source waters (Krishfield et al., 2014; Davis et al., 2014; Wang et al., 2019c)." (lines 31-32)

- You results are from a model and as such it would be useful to provide an extended discussion and assessment of the main differences with the real world. In terms of sea ice you assess your model globally but a lot of your results are then shown on maps. How does your model differ from the observations in terms of concentration and thickness spatially (compare to satellite data). Same question for your model SSH vs satellite derived SSH (i.e. from Armitage CPOM dataset or equivalent).

We added the assessment of sea ice concentration, thickness and drift in supporting material as **Figures S1-S3.** (figures are not repeated in this reply letter)

"The trend and interannual variability in Arctic sea ice volume and summer sea ice extent over the last four decades are reasonably simulated in the model in comparison to observations and reanalysis, and the simulated sea ice concentration, thickness and drift also compare well with satellite observations (Figures S1 - S4, Schweiger et al., 2011; Lavergne et al., 2010, 2019; Fetterer et al., 2017; Hendricks and Ricker, 2019). This model configuration can also well reproduce the trend and variability of the Arctic sea surface height observed by satellites and tide gauges (Xiao et al., 2020) and the recent changes in Arctic freshwater content (Wang, 2021)." (Lines 83-88)

AS SSH simulated in the model has been assessed in a recent separate paper (Xiao et al., 2020), we add the citation to it in the paper. The spatial SSH pattern (Fig. RE-1) and the SSH variability and trend (Fig. RE-2) compared with satellite observations, and the SSH variability compared with tide gauge data (Table RE-1) taken from Xiao et al. (2020) are shown below.

Fig. 2 Comparison of the mean SSH for the period 1993–2012 (a, b) and 2003–2014 (d, e). From left to right are a, d FESOM results, b DTU and e Armitage et al. (2016) observations, and c, f the residual between FESOM and observations

Fig. RE-1: Model assessment of SSH spatial patterns (Fig 2 of Xiao et al. 2020).